# Current-sensitive Hall effect in a chiral-orbital-current state

Yu Zhang[1,6], Yifei Ni [1,6], Pedro Schlottmann [2], Rahul Nandkishore[1,3], Lance E. DeLong [4] & Gang Cao [1,5] ✉

Chiral orbital currents (COC) underpin a novel colossal magnetoresistance in ferrimagnetic $Mn_3Si_2Te_6$. Here we report the Hall effect in the COC state which exhibits the following unprecedented features: (1) A sharp, current-sensitive peak in the magnetic field dependence of the Hall resistivity, and (2) A current-sensitive scaling relation between the Hall conductivity $\sigma_{xy}$ and the longitudinal conductivity $\sigma_{xx}$, namely, $\sigma_{xy} \propto \sigma_{xx}^{\alpha}$ with $\alpha$ reaching up to 5, which is exceptionally large compared to $\alpha \leq 2$ typical of all solids. The novel Hall responses along with a current-sensitive carrier density and a large Hall angle of 15% point to a giant, current-sensitive Hall effect that is unique to the COC state. Here, we show that a magnetic field induced by the fully developed COC combines with the applied magnetic field to exert the greatly enhanced transverse force on charge carriers, which dictates the COC Hall responses.

Our recent study revealed chiral orbital currents (COC) in a colossal magnetoresistance (CMR) material, ferrimagnetic $Mn_3Si_2Te_6$ (Fig. 1a–c)[1]. CMR is conventionally dictated by a spin polarization that drastically reduces spin scattering and thus electric resistance and is insensitive to applied electric currents. However, the CMR in $Mn_3Si_2Te_6$ occurs only when a spin polarization is absent[2] and is unprecedentedly current-sensitive[1]. The intriguing phenomena are explained in terms of a state of intra-unit-cell, $ab$-plane chiral orbital currents or $ab$-plane COC that generate net $c$-axis orbital magnetic moments ($M_{COC}$) which couple with the simultaneously ferrimagnetically ordered Mn spins[1]. In essence, the COC circulate along the edges of $MnTe_6$ octahedra to underpin an astonishing $10^7$-CMR that occurs without a net magnetic polarization along the magnetic hard axis (Fig. 1f)[1–3]. Note that a COC state was initially proposed and investigated in studies of high-$T_C$ cuprates, and later other materials [[1], references therein].

Ferrimagnetic $Mn_3Si_2Te_6$ with trigonal symmetry (P-31c)[4–7] orders at a transition temperature $T_C = 78$ K, with the magnetic easy axis along the $a$ axis and the magnetic hard axis along the $c$ axis (Fig. 1f)[1–10]. There are two inequivalent Mn1 and Mn2 sites in the unit cell. The $MnTe_6$ octahedra form a honeycomb sublattice of Mn1 ions in the $ab$ plane (Fig. 1b), whereas the $MnTe_6$ octahedra form a triangular sublattice of Mn2 ions sandwiched between the honeycomb layers (Fig. 1a, b)[4].

A recent neutron diffraction study reveals a noncollinear magnetic structure with the magnetic space group C2'/c', where the Mn spins lie predominantly within the $ab$ plane but tilt toward the $c$ axis by ~10 degree in ambient conditions (Fig. 1c)[8], which simultaneously breaks mirror and time reversal symmetries[8,9]. Such a noncollinear magnetic structure is essential for the COC to form below $T_C$ [1, Methods]. The COC circulate on the edges of $MnTe_6$ octahedra but predominantly within the $ab$ plane (Fig. 1a, b), and therefore generate orbital moments $M_{COC}$ primarily oriented along the $c$ axis (Fig. 1a)[1]. (Although the orbital moments could interact with each other at $H = 0$, this causes no long-range order, likely due to thermal fluctuations already evident in previous studies[2,8]). The $M_{COC}$ is estimated to be on the order of 0.1 $\mu_B$[1], and is coupled with the Mn spins, which yields an unusual spin-orbit effect that produces a large anisotropy field of 13 T (note that the orbital angular momentum is zero for the $Mn^{2+}$ ($3d^5$) ion with a half-filled $3d$ shell)[1,2,4]. In the absence of a magnetic field $H \parallel c$ axis, the net circulation of the COC is zero since it can circulate both clockwise and counterclockwise (Fig. 1d). This results in disordered circulation domains that cause strong scattering and high resistance.

[1]Department of Physics, University of Colorado at Boulder, Boulder, CO 80309, USA. [2]Department of Physics, Florida State University, Tallahassee, FL 32306, USA. [3]Center for Theory of Quantum Matter, University of Colorado at Boulder, Boulder, CO 80309, USA. [4]Department of Physics and Astronomy, University of Kentucky, Lexington, KY 40506, USA. [5]Center for Experiments on Quantum Materials, University of Colorado at Boulder, Boulder, CO 80309, USA. [6]These authors contributed equally: Yu Zhang, Yifei Ni. ✉e-mail: gang.cao@colorado.edu

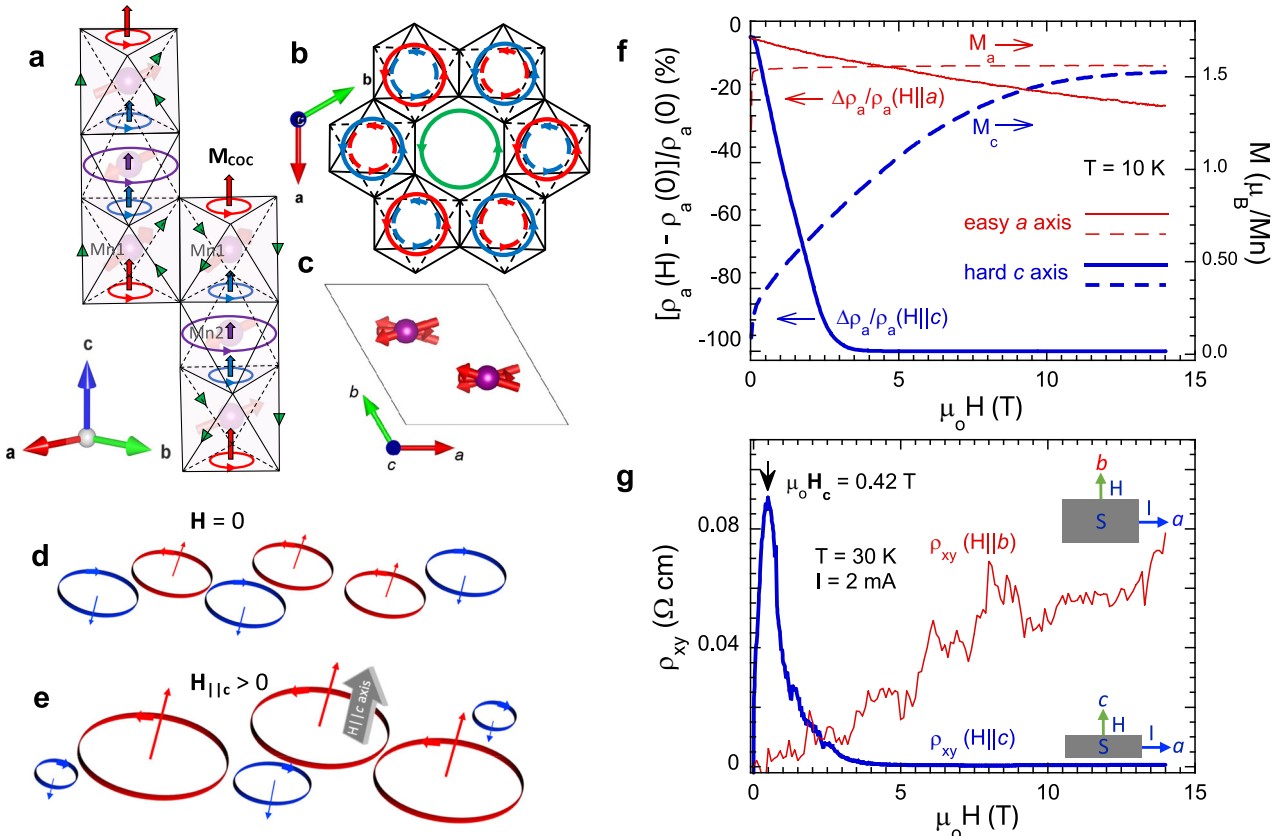

**Fig. 1 | Key structural and physical properties. a** The crystal and magnetic structure of $Mn_3Si_2Te_6$[1]. The colored circles and vertical arrows indicate the *ab*-plane COC and induced $M_{COC}$, respectively; different colors indicate different magnitudes of the *ab-plane* COC and $M_{COC}$; the green triangles denote *off-ab-plane* COC that are insignificant[1]; the faint cylindrical arrows are Mn spins. **b** The COC circulating in the honeycomb lattice in the *ab* plane[1]. **c** The canted Mn spins in the *ab* plane[8]. Schematics of the *ab*-plane COC at **H** = 0 (**d**) and $H_{||c}$ > 0 (**e**). **f** The magnetic field dependence of the *a*-axis magnetoresistance ratio $[\rho_{xx}(H)-\rho_{xx}(0)]/\rho_{xx}(0)$ and the magnetization M (dashed lines, right scale) for **H** || **c** axis (thick blue curve) and **H** || **a** axis (thin red curve)[2]. **g** The magnetic field dependence of the Hall resistivity $\rho_{xy}$ (**H** | **c**) (thick blue curve) and $\rho_{xy}$ (**H** | **b**) (thin red curve) at *T* = 30 K and *I* = 2 mA; the black arrow marks a critical field $H_C$. The insets are configurations for sample (S) measurements of $\rho_{xy}$ (**H** | **c**) (lower inset) and $\rho_{xy}$ (**H** | **b**) (upper inset).

However, application of **H** || **c** axis favors only one direction of circulation (i.e., either clockwise or counterclockwise) and expands its domains, concurrently reducing and eventually suppressing other domains with the opposite direction of circulation (Fig. 1e). The increased size of the preferred COC domains (and concurrent decrease in domain wall volume) leads to a sharp reduction in electron scattering, and thus the $10^7$-CMR (Fig. 1f)[1]. The COC as intrinsic currents are unusually susceptible to externally applied currents *I* that disrupt and eventually "melt" the COC state $\Psi_C$, resulting in a first-order transition to a trivial state $\Psi_T$ when *I* exceeds a critical threshold[1]. (Note that $\Psi_C$ refers to the COC state below $T_C$ at ambient conditions and the metallic state in the presence of **H** || **c** axis; $\Psi_T$ represents the trivial state above $T_C$ and a state where the COC are destroyed by applied currents[1]).

The interaction between COC and *I* presents new, intriguing physics that needs to be understood. We have applied the Hall effect as a fundamental, powerful probe of this interaction. There are a number of diverse models that have been formulated to explain the transverse conductivity $\sigma_{xy}$ of a variety of material types under various experimental conditions. The ordinary Hall effect (OHE) is attributed to the transverse emf proportional to **H** resulting from the Lorentz force on electrons. The OHE may be accompanied by an anomalous Hall effect (AHE) that is present in a ferromagnetic state with broken time-reversal symmetry[11–13]. The Hall resistivity $\rho_{xy}$ is thus anticipated to be proportional to the magnetization M[12–17]. More recent studies indicate that an intrinsic AHE (independent of scattering) can occur in a noncollinear

antiferromagnet with a strong spin-orbit interaction (SOI) so long as mirror and time reversal symmetries both are broken[18,19]. In other helical magnets such as MnSi, $\rho_{xy}$ exhibits an unusual stepwise field profile that is attributed to an effective magnetic field due to chiral spin textures[20]. Moreover, the Berry phase[13], which acts as an internal magnetic field[21,22], has been proposed as a source of a topological Hall effect (THE) observed in certain topological semimetals with a strong SOI [e.g.16–25].

Here, we show a strongly-current-sensitive Hall effect in ferrimagnetic $Mn_3Si_2Te_6$ that exhibits the following novel behaviors: (1) A distinct, sharp peak in the field dependence of $\rho_{xy}$ is a sensitive function of *I* (Fig. 1g) (so is the carrier density *n*) and (2) A scaling relation $\sigma_{xy} \propto \sigma_{xx}^{\alpha}$ is obeyed with $\alpha$-values ranging between 3 and 5, which are unprecedentedly large compared to $\alpha \leq 2$ typical of all solids[17], and sensitively depend on *I*. In addition, the Hall angle (given by the ratio of the Hall conductivity $\sigma_{xy}$ to the longitudinal conductivity $\sigma_{xx}$) reaches up to 0.15 which is comparable to values reported in magnets having a giant Hall effect[18,19,24,26]. An exceptionally large $\alpha$ indicates that in the COC state, $\sigma_{xy}$ rises with **H** much faster than $\sigma_{xx}$. We argue that the *c*-axis orbital moments $M_{COC}$ induced by the COC produce a real-space magnetic field $b_c$ that adds to an applied field **H** || **c** axis, i.e., **H** + $b_c$; as such the charge carriers gain an additional transverse velocity that generates the giant, current-sensitive Hall effect. This current-sensitive Hall effect shows no simple correlation with the magnetization M, or resemblance to conventional AHE (Fig. 1f, g), as predicted by the Karplus-Luttinger theory[12], nor does it behave as observed or expected

in other materials[18]. This unique Hall effect is a clear manifestation of the existence of the COC state and an intriguing interaction between the intrinsic and extrinsic currents.

Experimental details, including measurement techniques and processes, and additional data are described in Methods and Supplementary Figs. 1–8. All data reported here are reproduced in a dozen different samples (average sample size = $1.0 \times 1.0 \times 0.3$ mm$^3$). Note that Joule heating is inconsequential as it is discussed in[1] and confirmed by additional measurements specifically designed to investigate Joule heating in this material [Methods and Supplementary Figs. 4, 5].

## Results

### Magnetic field dependence of current-sensitive Hall resistivity

We first focus on $\rho_{xy}$ as a function of $\mathbf{H} \parallel c$ axis at $T = 30$ K as an example (Fig. 2). Note that $\rho_{xy}$ exhibits a sharp peak at a critical field $\mathbf{H_C}$ that marks an onset of the COC state $\Psi_C$. The peak, a hallmark of the COC state, is then followed by a rapid decrease of $\rho_{xy}$ by up to two orders of magnitude (Fig. 2a–e). (Note that $I = 1$ mA corresponds to a modest current density $J \approx 1$ A/cm$^2$ in the samples measured; for clarity we use $I$ in the discussion.) The peak shifts to higher fields with increasing $I$, revealing a sharp switching at $I = 3$ mA and 4.5 mA (Fig. 2c, d) before evolving into a broader peak at $I = 5$ mA (Fig. 2e), which signals a vanishing COC state $\Psi_C$ and an emerging trivial state $\Psi_T$. This behavior indicates that the COC weaken as $I$ is increased, and thus stronger $\mathbf{H} \parallel c$ axis are required to offset the disruption of the COC state caused by $I$. The correlation between $I$ and $\mathbf{H_C}$ at T = 30 K is illustrated in Fig. 2f. The peak at $\mathbf{H_C}$ (an indicator for $\Psi_C$) can persist up to $T_C = 78$ K so long as $I$ is small (e.g., 1 mA; see Fig. 2g–i for selected temperatures $T$). (Note that a strong hysteresis in $\rho_{xy}$ and $\rho_{xx}$ is seen between $\mathbf{H}$ ramping up and down [see Supplementary Fig. 1], as well as in previous studies[1,2], consistent with the presence of COC domains discussed above.) However, larger $I$ exceeding a certain threshold value $I_C$ can readily suppress the COC and recover $\Psi_T$ even well below $T_C$; this happens, for example, at 30 K when $I \geq 5$ mA. In these cases, the field dependence of $\rho_{xy}$ exhibits a behavior similar to that at 100 K and $I = 1$ mA, which is a benchmark for $\Psi_T$ that is signaled by only a broad or suppressed peak in the field dependence of $\rho_{xy}$ (Fig. 2j).

The rapid decrease in $\rho_{xy}$ at H > $\mathbf{H_C}$ indicates a fully developed $\Psi_C$ that is therefore much more conductive, i.e., $\rho_{xy}$ ($\mathbf{H} > \mathbf{H_C}$) $\ll$ $\rho_{xy}$ ($\mathbf{H_C}$), provided $I \leq 3$ mA. $\rho_{xy}$ ($\mathbf{H} > \mathbf{H_C}$) increases slowly and linearly with $\mathbf{H}$ (see left scale in Fig. 2k). However, with increasing $I$, $\Psi_C$ weakens and eventually transitions to $\Psi_T$, and accordingly, the slope of $\rho_{xy}$ evolves from positive to negative (right scale in Fig. 2k). Remarkably, at low $T$ and increasing $I$, $n(I)$ decreases by 3 orders of magnitude with a sign change at $I > I_C$ (Fig. 2l–o) that marks a change of state. For example, at $T = 50$ K (70 K), $n(I)$ is on the order of $10^{25}$/m$^3$ ($10^{26}$/m$^3$) and remains positive in $\Psi_C$ ($I < 3$ mA) and becomes negative, -$10^{22}$/m$^3$ (-$10^{23}$/m$^3$) in $\Psi_T$ ($I > 3$ mA) (Fig. 2m, n). At T = 100 K > $T_C$, n($I$) stays negative (Fig. 2o). These observations allow us to conclude that the charge carriers are primarily holes in $\Psi_C$ and electrons in $\Psi_T$ between $T_C$ and 120 K [Methods].

In sharp contrast to $\rho_{xy}$ for $\mathbf{H} \parallel c$ axis, $\rho_{xy}$ for $\mathbf{H} \parallel b$ axis exhibits a field dependence that is more consistent with an ordinary Hall effect (e.g., Fig. 1g and Supplementary Fig. 3) with no discernible evidence of an AHE, despite the fact that the $a$ axis is the magnetic easy axis, and the $a$-axis magnetization $M_a$ is fully saturated, reaching 1.6 $\mu_B$/Mn at

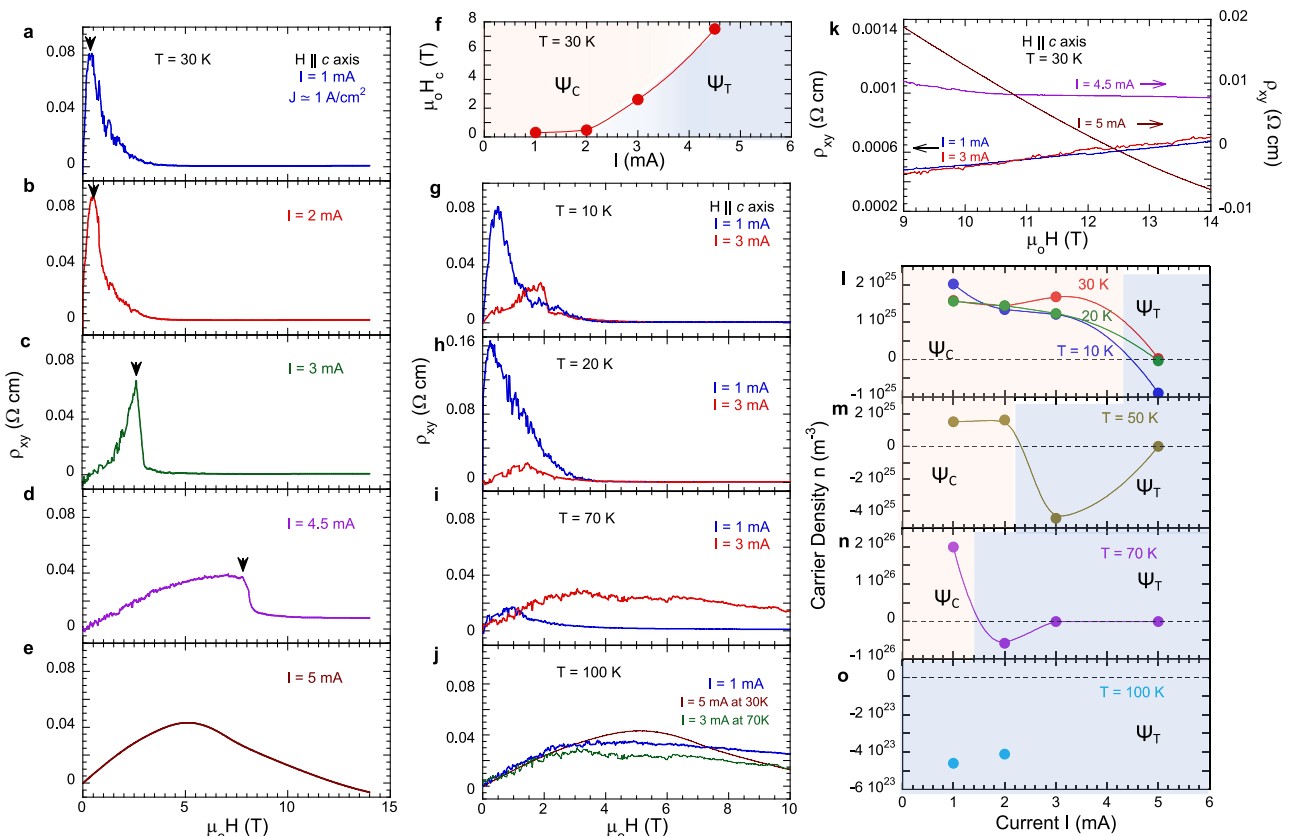

**Fig. 2 | The Hall effect as functions of magnetic field and external current. a–e** The magnetic field dependence of $\rho_{xy}$ ($\mathbf{H} \parallel c$) at 30 K for selected currents $I$; the black arrows mark the critical field $\mathbf{H_C}$. **f** The correlation between $\mathbf{H_C}$ and $I$ at $T = 30$ K; **g–j** The magnetic field dependence of $\rho_{xy}$ ($\mathbf{H} \parallel c$) at selected temperatures and currents. **k** The zoomed-in $\rho_{xy}$ ($\mathbf{H} \parallel c$) at T = 30 K (**a–e**) in a higher-field regime of 9-14 T. **l–o** The carrier density $n$ estimated from the data of $\rho_{xy}$ ($\mathbf{H} \parallel c$) for selected temperatures; the yellow and gray shaded areas are the COC state $\Psi_C$ and trivial states $\Psi_T$, respectively. Note that at $I = 5$ mA, $n = -9.9 \times 10^{22}$/m$^3$ and $-3.7 \times 10^{23}$/m$^3$ A at $T = 50$ and 70 K, respectively.

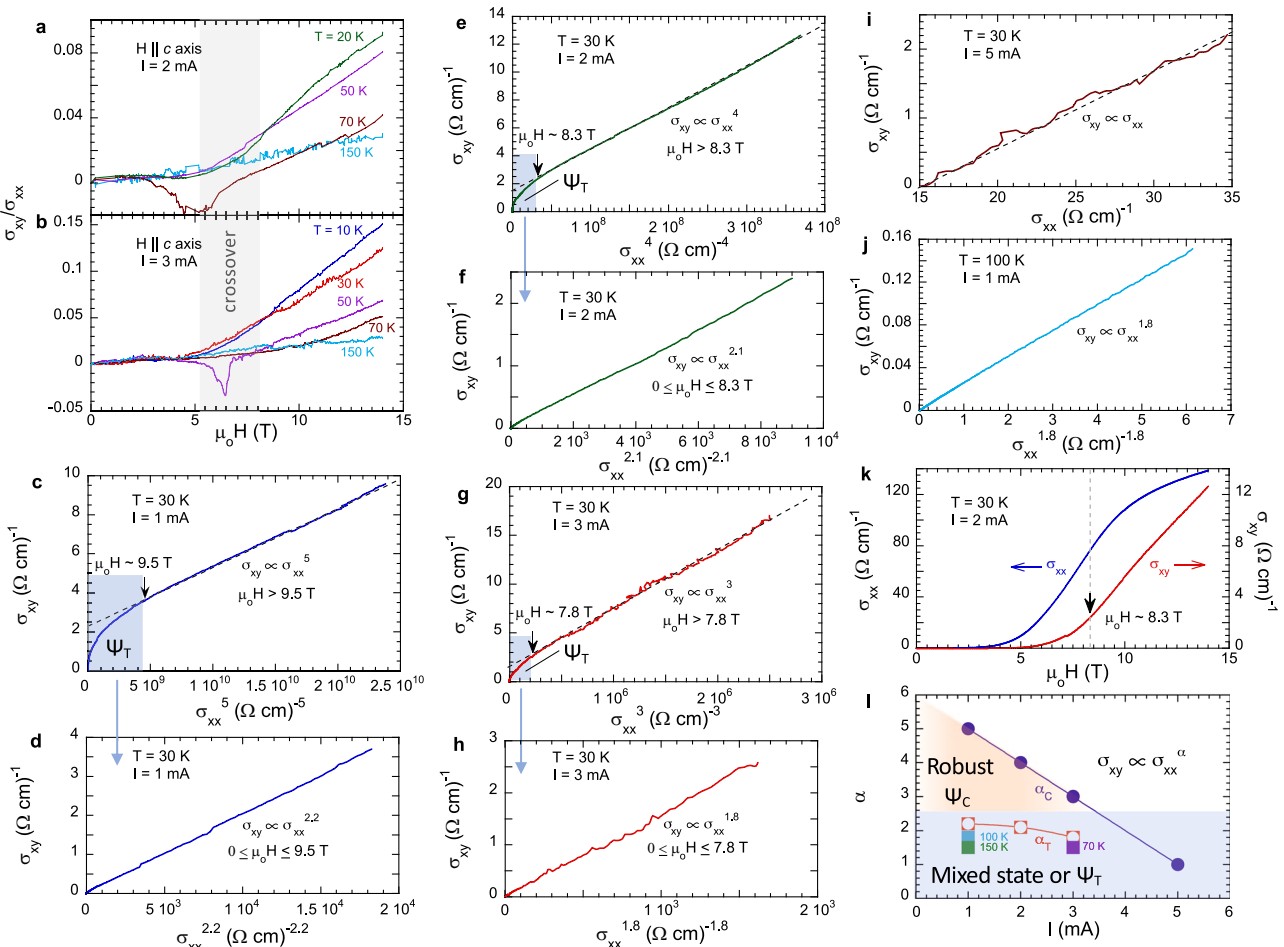

**Fig. 3 | The Hall angle and scaling relation as a function external current.** The magnetic field dependence of the Hall angle $\sigma_{xy}/\sigma_{xx}$ at $I = 2$ mA (**a**) and $I = 3$ mA (**b**) for selected temperatures; the gray band marks a crossover region between $\Psi_C$ and $\Psi_T$. Note that the gray area is approximately defined, and mainly serves as a guide to the eye. **c–i** The scaling relation $\sigma_{xy} \propto \sigma_{xx}^{\alpha}$ at $T = 30$ K for selected currents $I$. The shaded areas indicate a regime of $\Psi_T$ having a lower value of the exponent $\alpha$; the arrows indicate the cutoff field. **j** The scaling relation $\sigma_{xy} \propto \sigma_{xx}^{1.8}$ at $T = 100$ K and

$I = 1$ mA. **k** $\sigma_{xx}$ (left scale) and $\sigma_{xy}$ (right scale) as a function of **H** ∥ **c** axis at $T = 30$ K and $I = 2$ mA; the marked field 8.3 T indicates a crossover field above which $\sigma_{xy}$ rises much faster than $\sigma_{xx}$. **l** The phase diagram of the exponent $\alpha$ as a function of external current $I$. Note that $\Psi_C$ features $3 \leq \alpha = \alpha_{HF} < 5$ (yellow shaded area) whereas $\Psi_T$ $1 \leq \alpha = \alpha_{LF} < 2.2$ (gray shaded area). Note that $\alpha_c$ corresponds that for $\Psi_C$, and $\alpha_T$ for $\Psi_T$.

$\mu_o$**H** < 0.1 T (Fig. 1c)[1,2]. The corresponding $n(I)$ is on the order of $10^{23}$/m$^3$, comparable to that for $\Psi_T$ with **H** ∥ **c** axis (Fig. 2o). The contrasting Hall responses observed for **H** ∥ **c** axis and **H** ∥ **a** axis signal a highly anisotropic band structure[10], and more generally the novelty of the observed Hall effect.

### Current-sensitive scaling relation $\sigma_{xy} \propto \sigma_{xx}^{\alpha}$ and Hall angle

Moreover, for **H** ∥ **c** axis and $T < T_C$, the Hall angle, defined as the ratio of the Hall conductivity $\sigma_{xy} \left( = \frac{\rho_{xy}}{\rho_{xx}^2 + \rho_{xy}^2} \right)$ to the longitudinal conductivity $\sigma_{xx} \left( = \frac{\rho_{xx}}{\rho_{xx}^2 + \rho_{xy}^2} \right)$, or $\sigma_{xy}/\sigma_{xx}$, rises drastically when **H** enters a certain crossover region. As shown in Fig. 3a, b, $\sigma_{xy}/\sigma_{xx}$ initially increases slowly with **H** and remains smaller than 0.01 below 9 T. However, above the crossover region marked by the gray band in Fig. 3a, b, $\sigma_{xy}/\sigma_{xx}$ rapidly rises, reaching up to 0.15, which is among large values reported in magnets having a giant Hall effect[18,19,24,26]; the estimated mobility of charge carriers $\mu$ is on the order of 100 cm$^2$/V·s [Methods], one order of magnitude smaller than those of the forementioned magnets [e.g.[19,26]].

The large values of $\sigma_{xy}/\sigma_{xx}$ indicates that $\sigma_{xy}$ increases much faster than $\sigma_{xx}$ (~$\mu$) with increasing **H** when the COC state is fully developed in higher fields (> 6 T), giving rise to the further enhanced **M$_{COC}$** or **b$_c$**

that produces the additional transvers velocity of charge carriers. With vanishing $\Psi_C$ at $T > T_C$, $\sigma_{xy}/\sigma_{xx}$ expectedly rises with **H** only slightly (Fig. 3a, b). Remarkably, near the crossover region, $\sigma_{xy}/\sigma_{xx}$ exhibits a brief, yet prominent inverted peak at $I = 2$ mA when $T = 70$ K (Fig. 3a) and at $I = 3$ mA when T = 50 K (Fig. 3b). This peak persistently occurs whenever the system approaches the vicinity of the transition between $\Psi_C$ and $\Psi_T$ and is discussed further below.

We now examine the behavior of the scaling relation $\sigma_{xy} \propto \sigma_{xx}^{\alpha}$ for a few representative $I$ and $T$. Below $T_C$, $\sigma_{xy}$ scales with $\sigma_{xx}$ and generates two different values of the exponent, namely $\alpha_{HF}$ (obtained at higher fields) and $\alpha_{LF}$ (obtained at lower fields), which define two distinct regions corresponding to a fully developed $\Psi_C$ state and $\Psi_T$ or a mixed state of $\Psi_C$ and $\Psi_T$, respectively. A cutoff field that separates $\alpha_{HF}$ and $\alpha_{LF}$ falls in the crossover region marked in Fig. 3a, 3b. (Note that the gray area is approximately defined, and mainly serves as a guide to the eye.) An unanticipated, novel feature of this scaling relation is that the exponent $\alpha_{HF}$ is both unprecedentedly large and sensitive to $I$. Specifically, $\alpha_{HF}$ reaches 5 at $I = 1$ mA, as shown in Fig. 3c where the data above the cutoff field (9.5 T) perfectly trace the scaling relation $\sigma_{xy} \propto \sigma_{xx}^5$. With increasing $I$, $\alpha_{HF}$ reduces to 4 and 3 at $I = 2$ mA and 3 mA, respectively (Fig. 3e, g), suggesting that $\Psi_C$ gets weakened with increasing $I$. Below the cutoff field (shaded regions in Fig. 3c, e, g),

$\sigma_{xy}$ conforms to a scaling relation where $\alpha = \alpha_{LF} = 2.2$, 2.1 and 1.8 at $I = 1$ mA, 2 mA and 3 mA, respectively (Figs. 3d, f, h), indicating a vanishing $\Psi_C$ and an emerging $\Psi_T$. Applying $I = 5$ mA suppresses $\Psi_C$ and generates a linear scaling relation $\sigma_{xy} \propto \sigma_{xx}$ at 30 K (Fig. 3i). Similarly, a scaling relation with $\alpha_{LF} \leq 2$ is seen at $T = 70$ K and $I = 3$ mA [Supplementary Fig. 8], and $T = 100$ K and $I = 1$ mA (Fig. 3j). Importantly, $\sigma_{xy}$ and $\sigma_{xx}$ exhibit distinct **H** dependences at higher **H**. For example, at $I = 2$ mA and $T = 30$ K, $\sigma_{xy}$ rises rapidly and linearly whereas $\sigma_{xx}$ exhibits a tendency of saturation above **H** > 8.3 T (Fig. 3k); therefore, the increase of $\sigma_{xy}$ or $[\sigma_{xy}(14\,\text{T}) - \sigma_{xy}(8.3\,\text{T})]/\sigma_{xy}(8.3\,\text{T}) = 440\%$ above 8.3 T, but the value for $\sigma_{xx}$ is merely 83%. This implies that an additional driving force strongly affects the transverse current (discussed below) and explains the unusual scaling relation with the exceptionally large $\alpha$.

A phase diagram generated from the data illustrates that a hallmark of a fully developed $\Psi_C$ is a strongly current-dependent scaling relation $\sigma_{xy} \propto \sigma_{xx}^{\alpha}$ with an unprecedented range of $3 \leq \alpha = \alpha_{HF} \leq 5$ (Fig. 3l). (Note that $\alpha_c$ corresponds that for $\Psi_C$, and $a_T$ for $\Psi_T$.) However, when $\Psi_C$ is less robust or suppressed, this unique scaling relation is supplanted by another with $1 \leq \alpha = \alpha_{LF} < 2.2$, which is qualitatively similar to the range $1.6 \leq \alpha \leq 2$ commonly observed in insulators and bad metals having strong disorder[17,25].

### Temperature dependence of current-sensitive Hall resistivity

We now turn to the Hall effect as a function of $T$. Both $\rho_{xx}$ and $\rho_{xy}$ peak at $T = T_P$ well above $T_C$ (marked by hollow and solid arrows in Fig. 4a, b). The peak at $T_P$ is due to the broadening of the ferrimagnetic transition[2] by **H** (in this case, $\mu_o H_{\parallel c} = 7$ T). Both $T_P$ and $T_C$ progressively shift to lower $T$ with increasing $I$, but the temperature difference, $\Delta T$, remains essentially unchanged, i.e., $\Delta T = T_P - T_C \approx 32$ K at 7 T. As T decreases, $\rho_{xx}$ drops rapidly over the $\Delta T$ interval and reaches its lowest value slightly below $T_C$ (Fig. 4a). (Note that at $I \geq 5$ mA, $T_C$ is suppressed, thus $\Psi_T$ emerges.) On the other hand, $\rho_{xy}$ at $I = 1$ mA drops more than one order of magnitude from $T_P$ to $T_C$ (blue curve in Fig. 4b). With increasing $I$, $\rho_{xy}$ undergoes a rapid sign change from positive to negative to positive again over the span of $\Delta T$. This change results in a sharp inverted peak that progressively amplifies as $I$ increases (Fig. 4b). This peak could be a consequence of a transient crystal and/or band structure change

driven by $I$, which could strongly affect the COC[27–29]. However, the original crystal and/or band structure can be quickly recovered via either further increasing **H** or decreasing $T$ (see Fig. 3a, b and Fig. 4b).

## Discussion

While the microscopic origin of the observed Hall effect is yet to be established, we argue that the $c$-axis orbital moments $\mathbf{M_{COC}}$ induced by the COC play an essential role in this Hall effect. The high sensitivity of the observed Hall effect to small $I$ suggests a very delicate nature of the COC circulating along the edges of MnTe$_6$ octahedra. As already recognized[1], application of **H** || $\boldsymbol{c}$ axis expands the $ab$-plane COC domains with one direction of circulation and concurrently shrinks the COC domains with the opposite direction of circulation (Fig. 1d, e). The expanded $ab$-plane COC domains in turn generate stronger $c$-axis $\mathbf{M_{COC}}$, which render a magnetic field $\mathbf{b_c}$ aligned along the $c$ axis and proportional to **H**. This induced field $\mathbf{b_c}$ couples with **H** || $\boldsymbol{c}$ axis to yield an enhanced effective magnetic field $\mathbf{H + b_c}$ acting on itinerant holes in the COC state. The itinerant holes are strongly deflected by $\mathbf{H + b_c}$ and thereby gain a significant, additional transverse velocity, which is reflected by the greatly enhanced Hall current/conductivity schematically illustrated in Fig. 4c. This scenario qualitatively explains the key observations of this study. The sharp peak at H$_C$ in the field dependence of $\rho_{xy}$ signals the emergence of $\mathbf{b_c}$, and the rapid decrease in $\rho_{xy}$ at H > H$_C$ is a consequence of an added Hall current generated by $\mathbf{H + b_c}$ as $\Psi_C$ is fully developed (Fig. 2). When higher $I$ is applied, a stronger H$_C$ is needed (Fig. 2f) to offset the damage done to the COC in order to stabilize or further enhance the COC domains, thus $\mathbf{b_c}$. The unprecedentedly large $\alpha$ in the current-sensitive scaling relation $\sigma_{xy} \propto \sigma_{xx}^{\alpha}$ (Fig. 3) along with the large Hall angle can be ascribed to the additional transverse velocity of the holes ($\propto \mathbf{H + b_c}$) that drives an extraordinarily strong increase in $\sigma_{xy}$ in the high-field regime where $\mathbf{b_c}$ ($\propto$ H) gets further strengthened; in contrast, $\sigma_{xx}$ (~ $\mu$) in this high-field regime tends to saturate (Fig. 3k). This explains that the scaling relation with $3 \leq \alpha = \alpha_{HF} \leq 5$ is operative at much higher **H** only when the COC are fully established. In summary, the current-sensitive Hall effect is a novel transport phenomenon with great fundamental and technological promise, and merits extensive future investigations.

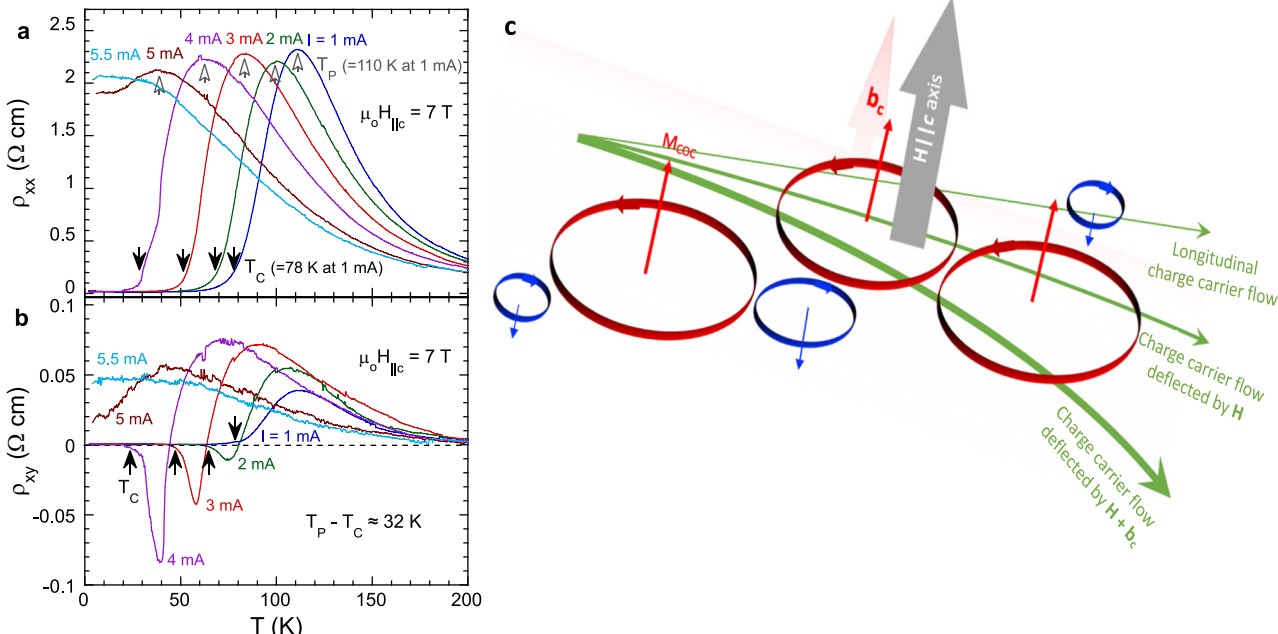

**Fig. 4 | The Hall effect and resistivity as functions of temperature and external current.** The temperature dependence of (**a**), $\rho_{xx}$ and (**b**), $\rho_{xy}$ at $\mu_o \mathbf{H}_{\parallel c} = 7$ T for selected currents $I$; the hollow and solid arrows mark the peak temperature $T_P$ and the Curie temperature $T_C$, respectively. $T_P - T_C \approx 32$ K for all $I$. **c** The schematic of the COC Hall effect.

## Methods

### Experimental details and data processing

The Hall resistivity is measured with a Quantum Design DynaCool 14 T PPMS. A standard 4-wire configuration for the Hall coefficient measurements is adopted. The applied current is supplied by a Keithley 6220 precision current source, which is paired with a Keithley 2812 A nanovoltmeter to measure the Hall voltage using the delta mode.

The measured voltage, $V_{meas}$, is a superposition of the Hall voltage, $V_{xy}$, and the longitudinal voltage, $V_{xx}$, due to the inevitably misaligned voltage leads (similarly the measured resistivity $\rho_{meas}$ is a superposition of $\rho_{xy}$ and $\rho_{xx}$). To eliminate $V_{xx}$, a field H sweep is performed and $V_{xy}$ is determined from the antisymmetrization of $V_{meas}$:

$$V_{xy}(\mathbf{H}) = \frac{1}{2}\left[V_{meas}(\mathbf{H}) - V_{meas}(-\mathbf{H})\right] \qquad (1)$$

Similarly, $V_{xx}$ is determined from the symmetrization of $V_{meas}$:

$$V_{xx}(\mathbf{H}) = \frac{1}{2}\left[V_{meas}(\mathbf{H}) + V_{meas}(-\mathbf{H})\right] \qquad (2)$$

Because of the nature of the COC domains in $Mn_3Si_2Te_6$, a hysteresis in the Hall measurements is observed between the field ramping up data and the field ramping down data (Supplementary Fig. 1). To eliminate the effect of hysteresis in the antisymmetrization of $V_{meas}$, we process the data in the following way: The data as a function of $\mathbf{H}$ are grouped into 4 parts: 14 T → 0 T (positive $\mathbf{H}$, |H| decreasing), 0 T → -14 T (negative $\mathbf{H}$, |H| increasing), -14 T → 0 T (negative $\mathbf{H}$, |H| decreasing), and 0 T → 14 T (positive $\mathbf{H}$, |H| increasing). The two sets of |H| decreasing data and the two sets of |H| increasing data are regrouped and $V_{xy}$ is obtained from the antisymmetrization of either group. The Hall voltage $V_{xy}$ retrieved this way is proven consistent, independent of the hysteresis.

The Hall voltage $V_{xy}$ is then normalized to the Hall resistivity $\rho_{xy} = V_{xy} \cdot t / I$, where $I$ is the current and $t$ is the thickness of the sample. The longitudinal resistivity $\rho_{xx}$ is determined in a similar way. The longitudinal conductivity $\sigma_{xx}$ and the Hall conductivity $\sigma_{xy}$ are thus given by $\sigma_{xx} = \rho_{xx}/(\rho_{xx}^2 + \rho_{xy}^2)$ and $\sigma_{xy} = \rho_{xy}/(\rho_{xx}^2 + \rho_{xy}^2)$.

### Noncolinear magnetic structures

Some topological Hall effect originates from noncollinear magnetic structures[18,19]; however, this type of Hall effect is never found to be highly sensitive to small currents. Moreover, the non-collinear magnetic structure, which persists up to high magnetic fields applied, is also essential for the formation of the COC[1]. Below $T_C$ (= 78 K), the ferrimagnetic order is observed with magnetic symmetry group C2'/c' (No. 15.89, BNS setting)[8]. This symmetry group allows certain configurations of Te orbital currents circulating within the unit cell. This symmetry-preserving COC state yields currents circulating on octahedral top/bottom faces. The resulting magnetic moments are all oriented exactly along the $c$ axis and can produce a nonzero net c-axis orbital moments $\mathbf{M_{COC}}$, which couple to the ferrimagnetic order due to Mn ions.

### Inconsequential Joule heating

Joule heating effects cause a continuous drift in local temperature. They are generally isotropic or diffusive and vary continuously with changing current. Such behavior is ruled out in the present study: The peak in the field dependence of $\rho_{xy}$ occurs and shifts abruptly with increasing external current $I$ (Fig. 2a−e). On the other hand, the peak in the field dependence of $\rho_{xy}$ at a constant current remains essentially unshifted with increasing temperature, as shown in Supplementary Fig. 2. The contrasting behaviors in Fig. 2a−e and Supplementary Fig. 2 are inconsistent with self-heating effects and rule out thereof. The absence of Joule heating is also confirmed in the data of the field

dependence of $\rho_{xy}$ at $\mathbf{H} \parallel \boldsymbol{b}$ axis, as shown in Supplementary Fig. 3, where the field dependence of $\rho_{xy}$ shows no significant change with increasing $I$.

Furthermore, we have conducted measurements to directly measure sample temperature T at different magnetic fields, using a Cernox thermometer attached to a single-crystal sample. As shown in Supplementary Fig. 4, the sample temperature is measured via the Cernox while a current is applied to the sample. We have applied the current up to 5 mA, which is the highest current used in our studies including the Hall effect study, and the magnetic field up to 14 T. Representative data are illustrated in Supplementary Fig. 5. For example, at $T = 30$ K, the sample temperature increase, $\Delta T$, can be up to 6 K at $I = 5$ mA and $\mathbf{H} = 0$; this value decreases to 3 K at 3 T and to 1-2 K at $\mathbf{H} > 3$ T. Similarly, $\Delta T$ at 5 mA is approximately 6 K at 10 K, and 2 K at 50 K and 70 K, as shown in Supplementary Fig. 5.

We would like to point out that all the COC phenomena reported in[1] and here occur at currents smaller than 3 mA. According to the data in Supplementary Fig. 5, the sample temperature increase is no more than 3 K or $\Delta T \leq 3$ K. In short, Joule heating does exist but causes no more than 6 K increase in sample temperature; its impact is therefore inconsequential for the phenomena reported here and[1].

### Roles of electric currents and electric field

From the data presented in Supplementary Fig. 6, 7, we can infer that the observed current-sensitive phenomena are primarily driven by electric current rather than electric field.

Our dielectric measurements using a Quad-Tech LCR meter reveal a large dissipation factor, DF, even at low temperatures, as shown in Supplementary Fig. 6. The large DF indicates a large leakage current, which therefore prevents a robust electric field from being established in this material, particularly at higher temperatures, say, 30 K at which most of our data is collected.

We have measured a dozen samples with varying sample size for this study. Compiling these data, we have found a converging behavior in these samples: The critical magnetic field $\mathbf{H_c}$ (defined by the peak that occurs in $\rho_{xy}$ ($\mathbf{H}$) (Fig. 2)) essentially follows the same current-density J dependence, as shown in Supplementary Fig. 7, which contains data obtained from four different samples. This strongly indicates that the Hall response depends on applied currents.

### Additional data for scaling relation

A scaling relation with $\alpha_{LF} \leq 2$ is observed at $T = 70$ K and $I = 3$ mA, as shown in Supplementary Fig. 8 This behavior provides additional evidence that an external current exceeding a critical current $I_C$ can suppresses the COC state below $T_C$ (= 78 K).

### Additional notes on the charge carriers

The charge carriers are primarily holes in $\Psi_C$ and electrons in $\Psi_T$ between $T_C$ and 120 K. Above 120 K, the charge carriers become holes again, as indicated by this and our previous studies[2].

### Estimate of mobility of charge carriers

$$\sigma_{xx} = e^2\tau n/m \sim 10^4 (\Omega m)^{-1}, \text{ and } n \sim 10^{25}/m^3 \text{ for } I \leq 3 \text{ mA}$$

$$\mu = e\tau/m = \sigma_{xx}/en \sim 10^{-2} \text{ m}^2/\text{Vs} = 100 \text{ cm}^2/\text{Vs}$$

## Data availability

The data that support the findings of this work are available from the corresponding authors upon request.

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

## Acknowledgements

This work is supported by National Science Foundation via Grant No. DMR 2204811.

## Author contributions

Y.Z conducted measurements of the physical properties and data analysis; Y.F.N. grew the single crystals, characterized the crystal structure of the crystals, and conducted measurements of the physical properties and data analysis. P.S. contributed to the theoretical analysis and paper revisions. R.N. conducted the theoretical analysis and contributed to paper revisions; L.E.D. contributed to the data analysis and paper revisions. G.C. initiated and directed this work, analyzed the data, constructed the figures, and wrote the paper.

## Competing interests

The authors declare no competing interests.
