## [Peer Review File · Nature Communications]

Reviewers' Comments:

Reviewer #1:

Remarks to the Author:

The work by Zhang et al. presents systematic Hall measurements on a highly intriguing system, revealing a Hall effect that depend sensitively on the current put through the material, with several other peculiarities. However, the manuscript has numerous scientific issues, and seemingly speculations speckled throughout. Therefore, I cannot recommend this work to be published, unless the issues described below can be satisfactorily addressed.

Main issues:

(1) I find the explanation that the CMR results from ordering of the CoC highly speculative. Can it really lead to a 107 CMR? Why are they not ordered at $H=0$ due to a coupling with the $\sim 10\%$ c-axis component of the Mn magnetic structure? How is the CoC CMR related to the CMR mechanism described in Ref. 3? Is it possible that a current somehow tunes the magnetic orientation/structure, which then leads to the experimental observations, without assuming a major role of the CoC? This does not seem unlikely, as the magnetic ordering temperature depends sensitively on I.

(2) Is the system a metal or an insulator, when the CoC are ordered, disordered, or destroyed? What roles do impurities play in realistic materials, which are being measured and reported in this work? Regarding Fig. 2l-2o, are the carriers intrinsic or due to impurities? Why do the carrier densities go to \sim zero in the Φ_T state for $I=5\text{mA}$? Are Φ_T and Φ_C intrinsically metallic or insulating states?

(3) Regarding Fig. 2k, it seems the authors assume the high-field response are due to the ordinary Hall effect. If so, it should be clearly stated. In this case, has the contribution from the conventional anomalous Hall effect ($\sim M$) been accounted for?

(4) The authors should NOT define a new Hall effect (COCHE), as the results can be broadly viewed as a type of conventional/unconventional anomalously Hall effect arising from inherent fields in the solid. The unique current response could be a result of the current modifying how the inherent moments are organized in the material, but the Hall effect itself does not defy current known examples (it seems like the conventional anomalous Hall effect, but with moments arising from CoC rather than typical magnetic moments).

(5) The hall effect in this compound could have many origins: the ordinary Hall, the conventional anomalous Hall, the unconventional anomalous Hall (or topological Hall) arising from topological band structures and magnetic structures. Since the system has a noncollinear magnetic structure at $H=0$, could it give rise to a scalar spin chirality when $H>0$, and thus a topological Hall effect? Since the system has a nodal line close to the Fermi level, could that also contribute to a topological Hall effect? Given the possible presence of these other Hall effects, how can they be ruled out in favor of a Hall effect from the CoC? Since it is in principle possible that a sizeable current modifies the magnetic/electronic structure of the material, which then lead to peculiar behaviors in the Hall effect via the aforementioned Hall effect mechanisms.

(6) Revise the manuscript in a way that ideally separate the experimental observations, and the possible mechanism proposed by the authors.

Minor issues:

(1) The paragraph introducing the Hall effect is very confusing. The conventional anomalous Hall effect is expected to scale with the magnetization, but there can also be unconventional anomalous Hall effects (or topological Hall effects) that result from scalar spin chirality or topological bands structures. The authors write in a way that seems to suggest the unconventional anomalous Hall effect (topological Hall effect) also scales with M . This paragraph needs to be clarified.

(2) The authors state H_C corresponds to the onset of Φ_C . How should this be understood, as Φ_C should be present already at $H=0$. What happens at H_C and why can it be used as an indicator for Φ_C ?

(3) The authors need to describe what Φ_C and Φ_T are exactly.

Reviewer #2:

Remarks to the Author:

In the present manuscript, Zhang et al. investigate the Hall resistance of $\text{Mn}_3\text{Si}_2\text{Te}_6$, highlighting its notable current dependence. This work appears to build upon the findings presented in the study published in Nature [Ref. 1], where the large magnetoresistance of $\text{Mn}_3\text{Si}_2\text{Te}_6$ and its current dependence were elucidated. The reported current values are merely on the order of 1 mA, which, if validated, could have significant implications.

Nevertheless, I have identified several areas of concern regarding the methodology and interpretations in this manuscript. It is crucial for the authors to address these issues comprehensively to ensure the robustness and significance of their findings. Until these matters are sufficiently resolved, I would hesitate to endorse this paper for publication.

(1) A primary concern that arises from the manuscript pertains to the potential effects of Joule heating, which becomes inevitably important when examining current-dependent properties of materials. The authors have referenced Ref. 1 to emphasize that Joule heating concerns have been completely addressed. To evaluate this claim, I further delved into the contents of Ref. 1.

Based on the data in Ref. 1, the resistivity of the studied material is approximately 100 ohm cm at 100K. When compared with standard metals, which have resistivities of order 100 uohm cm at the upper end, this material manifests a resistivity that is nearly a million-fold greater. Assuming an average sample size of 0.1 cm x 0.1 cm x 0.01 cm (given the manuscript omits specific dimensions), this equates to a resistance of 100 k Ω . Subjecting this to a current of 5mA, the ensuing heat generation approximates 2.5W. To put this into perspective, while a typical microwave oven operates at around 500W, 2.5W is colossal in condensed matter physics. From empirical knowledge, even 0.001W can elevate a sample's temperature by several Kelvins, making it implausible for measurements to remain unaffected by Joule heating.

While Ref. 1 posits several rationales to nullify concerns of Joule heating, they fall short of being universally convincing. The authors posit that heating-induced changes would be gradual and hence incompatible with the sharp transitions they observed. Nonetheless, it is plausible that in materials experiencing metal-insulator transitions, heating from electric currents might cause a sharp transition, particularly if a segment of the sample transforms into an insulator, consequently intensifying and hastening the heating. A parallel can be drawn to the transition from superconductors to metals under substantial currents. Further, while the authors emphasize resistance variances under different magnetic field orientations, it is not necessarily indicative of an absence of Joule heating, especially considering the pronounced resistance anisotropy the authors mention in different magnetic orientations. A few of their additional justifications can also be ascribed to the effects of Joule heating.

To allay the aforementioned reservations:

A direct temperature measurement of the sample can be employed. Techniques such as using a thermometer mounted directly to the sample within a PPMS setup (after evacuating the exchange gas) can yield accurate readings.

It would also be beneficial to discern whether the phenomena under scrutiny are influenced more by current or electric field. Conducting analogous experiments on samples of varied sizes can offer insights into this matter and potentially counteract heating concerns.

(2) The authors highlight that the Hall angle reaches a noteworthy value of 0.15. However, without a clear frame of reference, the significance of this value remains ambiguous. They compute the carrier density from the Hall resistivity and seemingly treat this resistance as ordinary Hall. It is worth noting, however, that an enlarged Hall angle for ordinary Hall resistance is not uncommon, in particular in materials with high carrier mobility.

Moreover, while the authors draw comparisons between the Hall angle in their study and those mentioned in Refs. 21-25, specifically papers 21-23 and 25 deal with the anomalous Hall effect. Such a comparison between the anomalous Hall effect and ordinary Hall resistance, seems

incongruous. Although a Hall angle of 0.15 might be considered substantial for an anomalous Hall effect, its magnitude does not stand out as particularly exceptional.

To further elucidate their findings, the authors should provide additional clarity on the significance of the reported Hall angle, especially in relation to its broader implications and context within the literature.

Minor Comments:

(1) The current symbol " I " should be rendered either in italics or bold to prevent misunderstanding.

(2) As highlighted earlier, it is pivotal for the authors to share the sample dimensions. This would enable readers to more effectively gauge potential heating issues.

Dear Reviewers,

We would like to extend our gratitude to each of you. Your recognition of the importance of this work is gratifying, and your insightful comments and suggestions have helped us significantly improve this manuscript.

With additional measurements and data, we have thoroughly addressed all concerns each of you raised, including those regarding the chiral orbital currents and Joule heating. We hope that you find them satisfactory.

In the following, we provide a summary of major changes to the manuscript based on the reports and our detailed point-by-point response to each of the two reports (see following pages).

Best wishes,

Gang

Gang Cao
Professor of Physics
University of Colorado at Boulder
On behalf of all authors

Summary of major changes to the manuscript and key points in our response

1. We have thoroughly revised the entire manuscript including Methods and Supplementary Figures to accommodate/address suggestions/comments of the Reviewers. All revisions are marked in red. For more clarity, the page numbers where revisions are made are listed in the end of relevant paragraphs in the following response.
2. We have revised the introduction to Hall effects (pages 3-4), as suggested by the Reviewer One.
3. In response to the Reviewer One's comments on our previous work in Ref.1, we have presented our discussion in the following several pages to address the robustness of the chiral orbital currents (COC) we report in Ref. 1 with new data and figures shown below (pages 3-11 of this response). In short, there is no existing/conventional model that can adequately explain a 10^7 -CMR that occurs without magnetic polarization and that is unprecedentedly sensitive to small currents. The COC model not only adequately explains

the experimental observations in Ref.1 and this manuscript but also is supported by our neutron work [Ref. 8].

4. We have revised the manuscript to further emphasize the unique, key features of the observed Hall effect, namely, the unprecedented current sensitivity of both the Hall effect and scaling relation $\sigma_{xy} \propto \sigma_{xx}^\alpha$ with α reaching to unprecedentedly large values (pages 1, 4, and 9). In the meantime, we have also significantly modified the statement on the Hall angle, including the estimated mobility of charge carriers (pages 1 and 6) at the Reviewer Two's suggestion. In addition, we have explicitly pointed out a current-sensitive carrier density $n(I)$ (pages 1, 4, and 6).
5. We are particularly mindful that Joule heating could cause spurious behavior and have extensively addressed the concern of Joule heating in both the main text (pages 4-5) and Methods (pages 13-14) with added Supplementary Figs. 4-5, which are obtained at the Reviewer Two's suggestion. In short, the increase in sample temperature due to small applied currents is no more than a few Kelvins; and these experimental observations are consistent with our estimated power due to Joule heating, which is merely on *the order of 1 mW* (see pages 16-22 of this response). Joule heating is therefore confirmed to be inconsequential in this material.
6. We have also added discussions in Methods with Supplementary Figs. 6-7 on the role of electric current and electric field. The added data indicate that no robust electric field can be established because our dielectric constant data show a large dissipation factor, DF. This discussion is also presented in response to the Reviewer Two's comments (pages 22-23 of this response).
7. We have added three new references to the manuscript, Refs. 21, 22, and 26.

Response (bold, *italic*) to comments/questions by the two reviewers

(References cited below, such as Ref. 1, are those cited in the manuscript)

Response to the Reviewer One

Reviewer #1 (Remarks to the Author):

Main issues:

(1) I find the explanation that the CMR results from ordering of the CoC highly speculative. Can it really lead to a 10^7 CMR? Why are they not ordered at $H=0$ due to a coupling with the $\sim 10\%$ c-axis component of the Mn magnetic structure? How is the CoC CMR related to the CMR mechanism described in Ref. 3? Is it possible that a current somehow tunes the magnetic orientation/structure, which then leads to the experimental observations, without assuming a major role of the CoC? This does not seem unlikely, as the magnetic ordering temperature depends sensitively on I .

We appreciate the question. In the following, we provide our detailed answer to this series of questions by addressing each separately.

1a. Current-sensitive 10^7 -CMR due to COC (I find the explanation that the CMR results from ordering of the CoC highly speculative. Can it really lead to a 10^7 CMR?)

Indeed, such a huge resistance reduction, in the absence of an apparent magnetic polarization, attests to the novelty of the current-sensitive CMR. For example, among many thought-provoking observations are:

- i. The 10^7 -CMR occurs only when magnetic polarization is avoided [Refs.1, 2, 3, 8].***
- ii. The 10^7 -CMR is extraordinarily sensitive to small currents (on the order of 1 mA) [Ref.1], which is unprecedented among all CMR reported [e.g., Rep. Prog. Phys. 69, 797 (2006)].***
- iii. The first-order bistable switching takes seconds or minutes to occur, in sharp contrast to picosecond time scales anticipated via electrons and/or Mn spins mechanisms [Ref.1].***

Our previous studies [Refs.1-2] considered all existing scenarios/models commonly applied to CMR (see a list below); but each of them requires a critical role played by magnetic polarization, and none of them explains the observed sensitivity to applied currents [Refs. 1-2]. All of these facts suggest a new model must be found.

For clarity, we list some existing scenarios/models for CMR cited in Refs 1-2:

- a. *A combined effect of double exchange, which dictates magnetism, and Jahn-Teller distortions, which drives electrical transport in the hole-doped manganites [Rep. Prog. Phys. 69, 797 (2006)].*
- b. *Phase separation in manganites [E. Dagotto, Nanoscale Phase Separation and Colossal Magnetoresistance (Berlin, Springer), 2002].*
- c. *Magnetic polarons with a sufficiently low carrier density in the ferromagnetic pyrochlore $Tl_2Mn_2O_7$ and $Eu_5In_2Sb_6$ [Phys. Rev. Lett. 81, 1314 (1998); npj Quantum Mater 5, 52 (2020)].*
- d. *The chiral anomaly $\vec{E} \cdot \vec{B}$ term that relies on parallel electrical currents and magnetic fields in topological Weyl/Dirac semimetals [Phys. Rev. B 88, 104412 (2013)].*

The following details our COC based model that adequately explains the observed CMR. Please note that COC have been implicated in a wide range of materials including cuprates, iridates, topological chalcogenides [see refs within Ref.1]. Our work in Ref.1 and this manuscript provide the first, direct evidence for interactions between intrinsic currents or COC and extrinsic currents or applied currents.

The COC, which couple to the ferrimagnetic state, originate from spontaneous breaking of time-reversal symmetry below the Curie temperature $T_c = 78$ K. The COC circulate along Te-Te edges of the $MnTe_6$ octahedra, producing orbital magnetic moments primarily oriented along the c-axis. Because of their symmetries, they necessarily entail an “antiferromagnetic” pattern of clockwise and counterclockwise COC circulations [Ref.1 Extended Data; also Fig.1 of the manuscript].

Fig.1. Schematic of COC and c-axis orbital moments.

Upper Panel:

Disordered circulation domains when $H_{||c} = 0$.

Lower Panel:

Ordered circulation domains when $H_{||c} > 0$.

In the absence of magnetic field applied along the c-axis, the net circulation of the COC is zero. This is because the COC are allowed to circulate both clockwise and counterclockwise, resulting

in domains with two opposite directions of the COC. Thermal fluctuations prevent the loops to order at $H=0$. Hence, the distribution of the signs of the loops is random. The disordered configuration of electron currents therefore gives strong scattering and high resistance below T_c (Fig. 1 Upper Panel). Applied magnetic fields along the c -axis which couple to the orbital magnetic moments only favor one COC circulation direction (either the counterclockwise or clockwise), and amplifies the COC with that direction, as well as increasing the local COC order parameter. The domains with the favored direction of the COC expand, and the domains with the opposite direction of the COC shrink (Fig. 1 Lower Panel). In short, application of $H \parallel c$ -axis drastically enhances the COC order parameter, therefore the hopping matrix elements, and accommodates more current passing through the sample, leading to the observed CMR.

This CMR effect can depend on details such as the structure of domain walls between opposite circulation domains and of the paths of net electron currents across COC circulation patterns, and thus is difficult to compute (see below); nevertheless, the combination of these mechanisms can account for the observed 7-order of magnitude reduction in resistivity.

The complications involved in a numerical estimate can be encapsulated in the following toy model. Let each Te-Te bond have a local COC vector order parameter δ that indicates the strength and direction of the electronic current on this bond in the ground state (which combine to specify the COC). We could also write a local "vector conductivity" σ , on this Te-Te bond, which is a function of the electronic current δ and of the (unknown) Hamiltonian H . Any partially-disordered configuration of the COC is described by the set of $\delta_{rr'}$, and gives rise to the set of conductivities $\sigma_{rr'}$. The network of vector conductivities could then be solved, as a kind of resistor network, to give a net conductivity across the sample. The resulting physical conductivity σ is a highly complicated function of the disordered distribution of local COC $\delta_{rr'}$ and of the Hamiltonian. Importantly, it is a highly sensitive function, so applying a magnetic field, which both increases the magnitude of δ and aligns the local COC circulations, is expected to have an enormous, sharp effect on σ , hence the observed CMR.

The fact that the CMR is so unusually sensitive to small, applied currents (≤ 2 mA) (see Fig. 2 of Ref. 1) or that the small currents are so highly detrimental to the CMR is an indication of a delicate interaction between the intrinsic COC and extrinsic DC currents. This interaction creates a nonequilibrium process that generates extraordinarily long-time scales. The bistable switching reported in Ref.1, is a manifestation of it. In essence, the time-dependent bistable switching is a result of the rigid coupling of the chiral orbital fluxes of the Te orbitals to the Te sublattice and the Mn spins, which is evidenced in the magnetostriction data in Fig.1c of Ref.1. This unique coupling enables the COC state to remain metastable over long time scales up to minutes, which are set by Te atomic motion and bond lengths or fluctuations/phonon effects

(rather than electrons or Mn spins), as we state in Ref.1. The rigidity of the coupling is related to the COC and the orbital magnetic moments.

As we point out in Ref. 1, our neutron study [Ref. 8] found that measured average magnetic moments for Mn1 and Mn2 are 4.55 and 4.20 μ_B , respectively, which are significantly smaller than 5 μ_B anticipated for Mn^{2+} ($3d^5$), suggesting that the measured moments may reflect a partial cancellation of the localized Mn spins by the c-axis COC moments. Accordingly, we can estimate an upper limit of the COC orbital moment to be on the order of 0.1 μ_B , as we also state in the manuscript.

We now turn to Figs. 1a-1b of the manuscript: Since moment = current x area, and considering the red, blue, and purple currents that generate c-axis components of the orbital magnetic moments (the green currents do not, thus are excluded in the estimate), an upper limit of the COC is estimated to be on the order of 0.1 mA/unit cell, which is huge value that could be affected by inaccurate estimates of other parameters. Nevertheless, we believe that it is reasonable because it is comparable to the applied critical currents, which are on the order of 1 mA (Ref.1 and the revised manuscript). This further explains the high susceptibility of the CMR and the Hall effect to small applied currents. The COC are microscopic currents that circulate within unit cells and may not dissipate. Indeed, the thermodynamic properties of this material show no particularly anomalous behavior in this aspect [e.g., our own studies of heat capacity [Ref.2] and neutron diffraction [Ref.8], and thermal conductivity, Seebeck effect and other thermal properties reported by others(Phys. Rev. B 98, 064423 (2018), Phys. Rev. B95, 174440 (2017), Phys. Rev. B 103, 245122 (2021), Phys. Rev. B 108, 054419 (2023)).

These values of the orbital moments and the COC, however rough, confirm an extraordinarily strong coupling of the orbital currents to the Te sublattice and the Mn spins, which dictates the COC state. In particular, the strong coupling sets time scales, which enables the COC state to remain metastable over seconds or minutes before undergoing a first-order transition to a normal state when applied DC currents exceed certain critical values on the order of 1 mA).

Such a long-time scale is unprecedented, but understandable. We infer that this transition mimics an ice-to-water phase transition. During ice melting, latent heat must be supplied to break the stiff hydrogen bonds, without raising temperature (it is a “thermal arrest”). The temperature of the system rises abruptly only when all hydrogen bonds are broken and the ice is completely melted. The analogy drawn here is that in the metastable state or during the COC melting, the applied DC currents (on the order of 1 mA) circulate in the sample to break COC in every unit cell, without causing a voltage or resistance increase before all COC are completely melted. This is because electrons always flow along the least resistive path, and this case, the

remaining COC. Breaking up the COC requires rearranging the Te orbitals, which alters lattice properties, which necessitates long delays in the switching. The analogy is completed by noting the larger the applied currents, the faster they can destroy all COC, which implies a shorter time delay.

In short, our COC picture satisfactorily explains the current-sensitive 10^7 -CMR. This model is further strengthened by the observed current-sensitive scaling relation, $\sigma_{xy} \propto \sigma_{xx}^\alpha$ with α ranging between 3 and 5 reported in this manuscript. Such a scaling relation is exceptionally large compared to $\alpha \leq 2$ typical of all known solids and follows no existing experimental or theoretical precedents.

1b. The relationship between Mn magnetic moments and CMR (Can it really lead to a 10^7 CMR? Why are they not ordered at $H=0$ due to a coupling with the $\sim 10\%$ c-axis component of the Mn magnetic structure?)

Besides the current sensitivity of the observed CMR, another key distinction of the observed CMR from other CMR is that it is inversely proportional to magnetic moments, as illustrated in Ref.1 and Fig. 1f of the revised manuscript. Here we present a more thorough examination of this matter.

Let's first examine the change of Mn magnetic moments as a function of applied magnetic field.

Our neutron studies of $Mn_3Si_2Te_6$ can be found in Ref. 8 [also cited in Ref.1] reveals a noncollinear magnetic structure below T_c where the moments lie predominantly within the basal plane but tilt toward the c axis by $\theta \approx 10^\circ$ in ambient conditions (see Inset in Fig.2 Left Panel). Application of magnetic field H along the c axis, the hard axis, renders a swift occurrence of CMR but only a slow tilting of the magnetic moments toward the c axis (see Fig.2 below) – The change of the magnetic moments (red dashed line in Left Panel and all curves in Right Panel of Fig. 2) does not at all track the change of the electrical resistivity ρ_{ab} (purple curve in Fig.2 Left Panel). Such an exceptional lack of correlation between the magnetization and magnetoresistance indicate that conventional models based on magnetic polarization alone cannot account for the observed CMR.

Fig. 2. Left Panel: Field dependence of the in-plane resistivity ρ_{ab} (purple) and canting angle or tilt angle θ (see inset) obtained from neutron diffraction and magnetization measurements. **Right Panel:** The tilt angle θ evolves with the applied magnetic field along the c axis up to 14 T for the undoped and Ge doped $\text{Mn}_3\text{Si}_2\text{Te}_6$ (Ref. 8). **Note that ρ_{ab} does not at all track θ .**

Moreover, Ge doping strongly enhances the CMR up to 9 orders of magnitude [Refs. 1 and 3] but, conversely, reduces the saturated magnetization M_s to $1.40 \mu_B/\text{Mn}$ in $\text{Mn}_3(\text{Si}_{1-x}\text{Ge}_x)_2\text{Te}_6$ from $1.60 \mu_B/\text{Mn}$ of the undoped sample, as shown in Fig. 3. It is also crucial that Ge doping expands the basal plane of the unit cell, as shown in the insert to Fig. 3 and Fig. 4. On the other hand, Se doping weakens the CMR by 3 orders of magnitude but enhances M_s to $1.7 \mu_B/\text{Mn}$ (Fig.3). Furthermore, Se doping shrinks the basal plane of the unit cell (see Fig. 3 inset and Fig.4). Such an inverse relationship between the CMR and M_s is unconventional but consistent with the COC argument: The stronger CMR is proportional to the COC-induced orbital moments. Since moment = current \times area, the orbital moments are stronger if the basal plane of the unit cell is larger, and weaker if the basal plane is smaller (Fig.3 inset and Fig.4). The larger 10^9 -CMR in the Ge doped sample with the expanded basal plane is related to the larger orbital moments, which in term results in a larger cancellation of the total magnetic moments, thus the smaller M_s ($=1.4 \mu_B/\text{Mn}$). Conversely, the smaller 10^4 -CMR in the Se doped sample with the shrunken basal plane is due to smaller orbital moments, which leads to a smaller cancellation of the total magnetic moments, hence the larger M_s ($= 1.7 \mu_B/\text{Mn}$). All this is consistent with our magnetostriction data, which confirms that applied magnetic fields H expand the basal plane of the unit cell when H is applied along the c axis but cause discernible shrinkage in the basal plane when H is along the a axis (see Fig. 5 below, which is Fig. 1c in Ref. 1). The magnetostriction is associated with the COC orbital moments because the Lorentz force acts to expand an ab-plane COC circulating on the rigid Te-Te edges of MnTe_6 octahedra when $H \parallel c$ -

axis. The net orbital moments, which equal the COC times the orbital area, i.e, moment = current \times area, naturally increases due to increasing the orbital area.

Fig.3. Left Panel: The magnetic field dependence of the a -axis magnetoresistance ratio defined by $\rho_a(H)/\rho_a(0)$ for $\text{Mn}_3(\text{Si}_{1-x}\text{Ge}_x)_2\text{Te}_6$ (red), $\text{Mn}_3\text{Si}_2(\text{Te}_{1-y}\text{Se}_y)_6$ (blue) and undoped compound (black). Inset: schematic illustration of the unit cell expansion and contraction due to Ge doping (red) and Se doping (blue), respectively. **Right Panel:** The magnetic field dependence of the ab -plane magnetization M_{ab} for the same samples. Note the inverse relationship between the CMR and M_{ab} .

Fig.5. Magnetic field dependence of the a -axis magnetostriction $\Delta a/a$ for H || c -axis (blue) and H || a -axis (red) for the undoped sample [Ref. 1].

Fig.4. Lattice parameters a and c axis as a function of dopant Ge (a) or Se (b) in $\text{Mn}_3(\text{Si}_{1-x}\text{Ge}_x)_2\text{Te}_6$ or $\text{Mn}_3\text{Si}_2(\text{Te}_{1-y}\text{Se}_y)_6$. Note that Ge doping (a) expands the unit cell, but Se doping (b) reduces the unit cell.

1c. Ref.3 (How is the CoC CMR related to the CMR mechanism described in Ref. 3?)

Let's now turn to Ref. 3. In Ref.3, the large anisotropy field of 13 T in the CMR is attributed to a topological nodal-line degeneracy of spin-polarized bands. In essence, an increasing c -axis spin magnetization lifts the energy of a particular band. The nodal line is preserved by the ab -plane spin magnetization, giving rise to the large anisotropy [Ref. 3]. However, like other CMR models, this model critically depends on the c -axis spin polarization, which is very weak and incompatible with the occurrence of the CMR, as discussed above and in Ref.1 and 8. Therefore, the mismatching responses of the Mn spins and the CMR to $H \parallel c$ axis (see Fig.2) reinforces an intriguing question as to why such an unusually small spin polarization could disproportionately result in the enormous 10^7 -CMR, which is also unusually current-sensitive. It is important to point out that the ferrimagnetic material in question is a highly correlated system with a sizable Sommerfeld coefficient of $\gamma = 23 \text{ mJ/mole K}^2$ [Refs. 1-2, 4-8], and therefore a model purely based on electronic bands with a band gap, as presented in Ref. 3, may not be adequate enough to explain the three key features, i, ii, and iii listed in the beginning of this response. The COC

driving the CMR requires strongly correlated physics that goes beyond simple modifications of a band structure due to interactions.

1d. Magnetic and crystal structures under applied DC currents and coupling of COC and ferrimagnetic state (Is it possible that a current somehow tunes the magnetic orientation/structure, which then leads to the experimental observations, without assuming a major role of the CoC? This does not seem unlikely, as the magnetic ordering temperature depends sensitively on I.)

Applied currents cause no discernible magnetic or crystal structural changes or phase transitions, according to our studies of neutron and x-ray diffraction [Ref. 8]. Application of 4 mA merely reduces the intensity of the magnetic (0,0,2) peak by 15% [Ref. 8]. Such small reduction in the magnetic peak intensity alone cannot account for the observed 10^7 -CMR. Temperature changes do not cause structural changes or transition as well (Fig. 6).

It is worth mentioning that the ferrimagnetic order is observed with magnetic symmetry group $C2'/c'$ (No. 15.89, BNS setting) [Ref. 8]. This symmetry group allows certain configurations of Te orbital currents circulating within the unit cell. This symmetry-preserving COC state yields currents circulating on octahedral top/bottom faces as well as around Mn2 sites and around the Si sites [Ref.1]. The resulting magnetic moments are all oriented exactly along the c-axis and can produce a nonzero net c-axis moments, which couple to the ferrimagnetic order due to Mn ions. This coupling generates an unusual spin-orbit effect that explains the large anisotropy field of 13 T mentioned above.

Fig. 6. No structural change below T_c (≈ 78 K) in $Mn_3Si_2Te_6$: Upper Panel: the honeycomb basal plane; Lower Panel: Snapshot of neutron diffraction pattern of the honeycomb lattice at $T = 50$ K.

Nevertheless, we value this question and have revised both the main text and Methods to clarify the Reviewer's concerns.

(2) Is the system a metal or an insulator, when the CoC are ordered, disordered, or destroyed? What roles do impurities play in realistic materials, which are being measured and reported in this work? Regarding Fig. 2l-2o, are the carriers intrinsic or due to impurities? Why do the carrier densities go to ~zero in the Phi_T state for I=5mA? Are Phi_T and Phi_C intrinsically metallic or insulating states?

We appreciate this question. Once again, we provide our answer to this question by separately addressing each of the sub-questions.

2a. The relationship between the COC and the electronic states (Is the system a metal or an insulator, when the CoC are ordered, disordered, or destroyed?)

As discussed above, this material is an insulator in the absence of $H \parallel c$ axis because the net circulation of the COC is zero, and disordered circulation domains cause strong scattering and high resistance (the resistivity is up to $10^7 \Omega\text{cm}$). In the presence of $H \parallel c$ axis, the COC domains with one circulation direction expand while the COC domains with the opposite circulation direction shrink, resulting in drastic reduction of scattering, thus the CMR [Ref. 1].

When applied currents exceed a critical current I_c on the order of 1 mA (Ref. 1), the CMR diminishes because the applied currents larger than I_c "melt" or destroy the COC. Consequently, the material becomes less metallic [Fig .2 of Ref. 1]. Therefore, the observed CMR in this material is extraordinarily current-sensitive.

In the revised manuscript, we have further clarified this [pages 2 - 3].

2b. The role of impurities and carrier density (What roles do impurities play in realistic materials, which are being measured and reported in this work? Regarding Fig. 2l-2o, are the carriers intrinsic or due to impurities? Why do the carrier densities go to ~zero in the Phi_T state for I=5mA? Are Phi_T and Phi_C intrinsically metallic or insulating states?)

The quality of single-crystal samples from which the data are collected is thoroughly characterized via neutron and x-ray diffraction and energy dispersion x-ray (EDX) [Refs. 1, 2, and 8]. No unwanted impurities are discerned. The data reported are highly reproducible (confirmed in a dozen samples from different sample batches; also see discussion below) and therefore intrinsic. However, intentionally doped impurities do profoundly impact the physical properties of this material. As discussed above (Fig. 3), Ge doping enhances the CMR by 2 orders of magnitude, but Se doping reduces it by 3 orders of magnitude, compared to that of the undoped sample [Refs.1 and 3]. This is primarily because the dopants alter the basal plane of

the unit cell, thus strongly affect the COC-induced orbital moments (= current x basal plane area), as pointed out above (Fig. 4). The carriers in Fig. 2l-2o are intrinsic and not due to impurities.

The carrier densities in question are finite, not zero, but 4 orders of magnitude smaller (on the order of 10^{22} or $10^{23}/\text{m}^3$) than those at smaller currents ($\sim 10^{25}$ - $10^{26}/\text{m}^3$); please see Fig. 7 below. Indeed, since the data cannot be presented in a logarithmic scale, they can be confusing. We have revised the text and captions to avoid the confusion (pages 6 and 11).

Fig. 7. Clarifications of Fig. 2 of the manuscript:

Please note that the data points at high currents are finite but 4 orders of magnitude smaller than those at smaller currents.

Ψ_C refers to the COC state below T_c ($= 78$ K). The COC state Ψ_C is a metallic state in the presence of $H \parallel c$ -axis; Ψ_T represents the trivial state above T_c or a state where the COC are destroyed by applied currents, thus an insulating state. We have revised the text to make it clearer (page 3).

(3) Regarding Fig. 2k, it seems the authors assume the high-field response are due to the ordinary Hall effect. If so, it should be clearly stated. In this case, has the contribution from the conventional anomalous Hall effect ($\sim M$) been accounted for?

We appreciate this question. The Hall response in question at high currents ($I > 3$ mA) is indeed mostly due to the ordinary Hall effect because the COC are destroyed by high currents. We have further clarified this point. No conventional anomalous Hall effect is discerned in this material (e.g., Fig. 1g of the manuscript and Supplementary Fig.3). We have explicitly stated it in the revised version (page.4).

(4) The authors should NOT define a new Hall effect (COCHE), as the results can be broadly viewed as a type of conventional/unconventional anomalously Hall effect arising from inherent fields in the solid. The unique current response could be a result of the current modifying how the inherent moments are organized in the material, but the Hall effect itself does not defy current known examples (it seems like the conventional anomalous Hall effect, but with moments arising from CoC rather than typical magnetic moments).

Taking the reviewer suggestion, we have removed the term COCHE. As discussed above, the observed Hall effect shows no sign of a conventional anomalous Hall effect despite the strong ab-plane magnetization (indeed, the CMR occurs only when the ab-plane magnetization is avoided (see Fig.1f-1g).

We believe the observed Hall effect in $Mn_3Si_2Te_6$ is indeed unique, for the following reasons: 1) No known anomalous Hall effect is current-sensitive. 2) No known anomalous Hall effect obeys a scaling relation $\sigma_{xy} \propto \sigma_{xx}^\alpha$ with α ranging between 3 and 5, as observed in $Mn_3Si_2Te_6$; whereas $\alpha \leq 2$ is typical of all solids previously studied.

(5) The hall effect in this compound could have many origins: the ordinary Hall, the conventional anomalous Hall, the unconventional anomalous Hall (or topological Hall) arising from topological band structures and magnetic structures. Since the system has a noncollinear magnetic structure at $H=0$, could it give rise to a scalar spin chirality when $H>0$, and thus a topological Hall effect? Since the system has a nodal line close to the Fermi level, could that also contribute to a topological Hall effect? Given the possible presence of these other Hall effects, how can they be ruled out in favor of a Hall effect from the CoC? Since it is in principle possible that a sizeable current modifies the magnetic/electronic structure of the material, which then lead to peculiar behaviors in the Hall effect via the aforementioned Hall effect mechanisms.

We appreciate this point. We have presented more arguments on this matter in the revised version, both the main text and Methods (pages 2 and 13)

On this point, we would like to iterate our discussion in our response 1c above regarding Ref. 3.

Below we summarize a few points discussed above which we think could rule out other established mechanisms:

- a. ***Topological nodal-line degeneracy of spin-polarized bands is unlikely. As pointed out above in Response to Question 1, like other CMR theories, this scenario critically depends on the c-axis Mn spin polarization, which is very weak and does not match the rapid occurrence of the CMR due to $H \parallel c$ axis (see Fig.2). Therefore, topological nodal-line degeneracy cannot explain the underlying question as to why such an unusually small spin polarization could disproportionately result in the enormous 10^7 -CMR, which***

is also unusually current-sensitive. Moreover, $\text{Mn}_3\text{Si}_2\text{Te}_6$ undergoes long-range magnetic order and is a highly correlated system; which is corroborated by a sizable Sommerfeld coefficient of $\gamma = 23 \text{ mJ/mole K}^2$ [Ref. 2]. Therefore, a picture solely based on electronic bands with a band gap, as proposed in Ref.3, cannot adequately explain the experimental observations, particularly considering the mismatches between the Mn polarization and the CMR [Fig.2]. The COC picture requires strongly correlated physics and indeed well explains the current-sensitive CMR without magnetic polarization [Ref. 1 and its Extended Data].

- b. The Reviewer's concern regarding non-collinear magnetic structures is appreciated. Indeed, such magnetic textures are known to underpin some topological Hall effects [18, 19]; however, this type of Hall effect is never observed to be highly sensitive to small currents. Moreover, the non-collinear magnetic structure, which persists up to high applied magnetic fields, is also essential for the formation of the COC. Below $T_c (= 78 \text{ K})$, the ferrimagnetic order is observed with a magnetic symmetry group $C2'/c'$ (No. 15.89, BNS setting) [Ref. 8]. This symmetry group permits certain configurations of Te orbital currents circulating within the unit cell. This symmetry-preserving COC state yields currents circulating on octahedral top/bottom faces. The resulting magnetic moments are all oriented along the c-axis and can produce a nonzero net c-axis orbital moments, which couple to the ferrimagnetic order due to Mn ions.*
- c. As already pointed out above, an unprecedented feature of the observed Hall response is the current-sensitive scaling relation in the COC state $\sigma_{xy} \propto \sigma_{xx}^\alpha$ with α ranging between 3 and 5 (Fig. 3 of the revised manuscript). This along with other Hall responses reported here manifests a novel Hall effect.*
- d. Given all above, it may not be surprising that no conventional Hall effects are discerned in this material.*

We have added the discussion on noncollinear magnetic structure to the Methods (page 13).

(6) Revise the manuscript in a way that ideally separate the experimental observations, and the possible mechanism proposed by the authors.

We have extensively revised our manuscript based on this suggestion.

Minor issues:

(1) The paragraph introducing the Hall effect is very confusing. The conventional anomalous Hall effect is expected to scale with the magnetization, but there can also be unconventional

anomalous Hall effects (or topological Hall effects) that result from scalar spin chirality or topological bands structures. The authors write in a way that seems to suggest the unconventional anomalous Hall effect (topological Hall effect) also scales with M . This paragraph needs to be clarified.

We have extensively revised the introduction of the manuscript to address this suggestion (pages 3 and 4).

(2) The authors state H_C corresponds to the onset of Φ_C . How should this be understood, as Φ_C should be present already at $H=0$. What happens at H_C and why can it be used as an indicator for Φ_C ?

(3) The authors need to describe what Φ_C and Φ_T are exactly.

We have revised the manuscript to take the above two suggestions into account. In essence, Ψ_C refers to the COC state extant below $T_c = 78$ K. The COC state Ψ_C in the presence of $H \parallel c$ -axis is a metallic state. Ψ_T represents the trivial state above T_c or a state in which the COC are destroyed by applied currents, resulting in an insulating state (page 3).

Response to the Reviewer Two

Reviewer #2 (Remarks to the Author):

In the present manuscript, Zhang et al. investigate the Hall resistance of $Mn_3Si_2Te_6$, highlighting its notable current dependence. This work appears to build upon the findings presented in the study published in Nature [Ref. 1], where the large magnetoresistance of $Mn_3Si_2Te_6$ and its current dependence were elucidated. The reported current values are merely on the order of 1 mA, which, if validated, could have significant implications.

Nevertheless, I have identified several areas of concern regarding the methodology and interpretations in this manuscript. It is crucial for the authors to address these issues comprehensively to ensure the robustness and significance of their findings. Until these matters are sufficiently resolved, I would hesitate to endorse this paper for publication.

The Reviewer's recognition of the importance of this work is gratifying, and his/her thoughtful reading and insightful comments and suggestions have helped us significantly improve this manuscript.

(1) A primary concern that arises from the manuscript pertains to the potential effects of Joule heating, which becomes inevitably important when examining current-dependent

properties of materials. The authors have referenced Ref. 1 to emphasize that Joule heating concerns have been completely addressed. To evaluate this claim, I further delved into the contents of Ref. 1.

Based on the data in Ref. 1, the resistivity of the studied material is approximately 100 ohm cm at 100K. When compared with standard metals, which have resistivities of order 100 uohm cm at the upper end, this material manifests a resistivity that is nearly a million-fold greater. Assuming an average sample size of 0.1 cm x 0.1 cm x 0.01 cm (given the manuscript omits specific dimensions), this equates to a resistance of 100 k Ω . Subjecting this to a current of 5mA, the ensuing heat generation approximates 2.5W. To put this into perspective, while a typical microwave oven operates at around 500W, 2.5W is colossal in condensed matter physics. From empirical knowledge, even 0.001W can elevate a sample's temperature by several Kelvins, making it implausible for measurements to remain unaffected by Joule heating.

While Ref. 1 posits several rationales to nullify concerns of Joule heating, they fall short of being universally convincing. The authors posit that heating-induced changes would be gradual and hence incompatible with the sharp transitions they observed. Nonetheless, it is plausible that in materials experiencing metal-insulator transitions, heating from electric currents might cause a sharp transition, particularly if a segment of the sample transforms into an insulator, consequently intensifying and hastening the heating. A parallel can be drawn to the transition from superconductors to metals under substantial currents. Further, while the authors emphasize resistance variances under different magnetic field orientations, it is not necessarily indicative of an absence of Joule heating, especially considering the pronounced resistance anisotropy the authors mention in different magnetic orientations. A few of their additional justifications can also be ascribed to the effects of Joule heating.

Fig.8. Representative single crystal image. The average size of the crystal samples studied is 1 mm x 1 mm x 0.3 mm.

We are particularly mindful that Joule heating could cause spurious behavior. The Reviewer's thoroughness is therefore very much appreciated. We should have included the average sample size in the manuscript, which is proximately 1 mm x 1 mm x 0.3 mm, close to the estimate by the Reviewer (see Fig. 8 and Fig. 11 illustrating a setup of the sample and a Cernox thermometer to perform the measurements the Reviewer Two suggested). Using this average sample size, we calculate the Joule heating as follows:

According to Fig. 1a of Ref. 1, at $I = 10 \text{ nA}$ and $H_{||c} = 0$ in the insulating state, the resistivity $\rho \sim 100 \text{ } \Omega \text{ cm}$ (and ρ decreases rapidly with increasing I);

the average cross-sectional area $A = 3 \times 10^{-3} \text{ cm}^2$;

the average separation between the 2 voltage leads $L = 10^{-2} \text{ cm}$;

thus, the average resistance $R = \rho L/A = 100 \text{ } \Omega \text{ cm} \times 10^{-2} \text{ cm} / 3 \times 10^{-3} \text{ cm}^2 = 330 \text{ } \Omega$.

Using $I = 5 \text{ mA}$, the largest current applied in this work,

the dissipated power $P = I^2 \times R = 25 \times 10^{-6} \text{ A}^2 \times 330 \text{ } \Omega = 8.25 \times 10^{-3} \text{ W} = 8.25 \text{ mW}$.

The order of magnitude of this value is consistent with that calculated based on the IV curves in Fig.9 [Fig. 3 of Ref. 1]. Here are two examples:

a. for $I = 2 \text{ mA}$, $V = 1.4 \text{ V}$ at 10 K , $P = IV = 2 \times 10^{-3} \text{ A} \times 1.4 \text{ V} = 2.8 \times 10^{-3} \text{ W} = 2.8 \text{ mW}$;

b. for $I = 20 \text{ mA}$, $V = 0.25 \text{ V}$ at 10 K , $P = 5.0 \text{ mW}$.

Note that the small input power further decreases with increasing T and H [Ref. 1].

Therefore, the Joule heating in question is on the order of 1 mW , 3 orders of magnitude smaller than that estimated in the report. But the Reviewer Two correctly estimates that Joule heating of milli-Watts can raise temperature by a few Kelvins. Following the Reviewer's suggestions, we have conducted a new, direct investigation of Joule heating in the material, which is discussed below.

Before discussing the new results on Joule heating, we would like to make a few observations of our data in Fig. 10 below [Fig. 4 of Ref. 1] in response to the Reviewer's concern:

Fig. 9. The a -axis I-V characteristic (a) At various temperatures and $H = 0$; (b) Details of the outlined area in (a); note the first-order transition at I_c [Ref.1].

a. Fig. 10a: In the absence of a magnetic field H , the bistable switching leads to a first-order jump to a much higher value of voltage V or resistance R with increasing I . If this switching were due to Joule heating, V or R would decrease rather than increase

because R at $H = 0$ follows an insulating behavior (i.e., the higher temperature corresponds to lower resistance) [Fig. 2 of Ref. 1]. Moreover, the switching is essentially between two values of V , thus the bistable switching basically independent of applied currents (this is particularly true in the data in Fig. 10b). Please also note that the switching or the jump in V , $\Delta V = 0.52$ V, is triggered by a tiny increase of applied I , $\Delta I = 0.005$ mA (= 2.035 – 2.030 mA).

- b. Fig. 10b: In the presence of $H \parallel c$ axis, the first-order bistable switching is drastically enhanced: A tiny current increase of $\Delta I = 0.01$ mA causes a first-order jump in V , $\Delta V = 0.99$ V, this is 2000% increase in V . Please also note there are only two voltage states, independent of I .
- c. Fig. 10c: However, if H is applied along the a -axis, the first-order bistable switching essentially disappears. The value of the voltage V now sensitively depends on the value of applied current I , consistent with the behavior anticipated for a normal state, and the Ohm behavior is nearly recovered. The gradual increase of V with time could be due to slight misalignments of magnetic field with the a axis.

Fig. 10. Time-dependent bistable switching: The a -axis voltage V_a as a function of time t at $T = 10$ K for (a) $H = 0$, (b) $\mu_0 H_{\parallel c} = 7$ T and (c) $\mu_0 H_{\parallel a} = 7$ T [Ref.1].

- d. All the behaviors, a through c above, are inconsistent with Joule heating. The point we would like to stress here is that Joule heating, which does exist here, is inconsequential for the main conclusions of our studies (more data and discussion below). We do agree with the Reviewer that the applied current could drive a transition, but not via Joule heating. This transition is between the COC and normal states, as discussed above as well as in Ref. 1 and its Extended Data. We infer that this transition may mimic an ice-to-water phase transition. During ice melting, the thermal energy is spent to break the stiff hydrogen bonds, without raising temperature. The temperature of the system rises abruptly only when all hydrogen bonds are broken, i.e., the ice is completely melted. An analogy drawn here is that at the metastable state or during the COC melting, the applied DC currents (on the order of 1 mA) circulate in the sample to break COC in every unit cell, without causing a voltage or resistance increase before all COC are completely melted. This is because electrons always flow along the least resistive path, and in this case, the remaining COC, according to the two-channel model. The melting COC causes rearrangements of Te orbitals and atoms, more generally, changing lattice properties, therefore a long delay for switching. The larger the applied currents, the faster they can destroy all COC, thus a shorter time delay.***

To allay the aforementioned reservations:

A direct temperature measurement of the sample can be employed. Techniques such as using a thermometer mounted directly to the sample within a PPMS setup (after evacuating the exchange gas) can yield accurate readings.

It would also be beneficial to discern whether the phenomena under scrutiny are influenced more by current or electric field. Conducting analogous experiments on samples of varied sizes can offer insights into this matter and potentially counteract heating concerns.

At Reviewer's suggestion, we have conducted the following measurements and analysis:

- a. Sample temperature as a function of applied current. Using a Cernox thermometer attached to a single-crystal sample, we have conducted measurements of sample temperature T at different magnetic fields. As shown in Fig. 11, the sample temperature is measured using the Cernox while a current is applied to the sample. We have applied the current up to 5 mA, which is the highest current used in our studies including the Hall effect study, and the magnetic field up to 14 T. Representative data are illustrated in Figs. 12-13. For example, at $T = 30$ K, the sample temperature increase, ΔT , can be up to 6 K at $I = 5$ mA and $H = 0$; this value decreases to 3 K at 3 T and to 1-2 K at $H > 3$ T (Fig. 12). Similarly, ΔT at 5 mA is approximately 6 K at 10 K, and 2 K at 50 K and 70 K, as shown in Fig. 13.***

b. We would like to point out that all the COC phenomena reported in Ref. 1 and in this manuscript occur at currents smaller than 3 mA. According to the data in Figs. 2 – 13, the sample temperature increase is no more than 3 K or $\Delta T \leq 3$ K.

Fig. 11. Joule heating measurements: A Cernox thermometer (gold, front) is thermally contacted with a single-crystal sample of $\text{Mn}_3\text{Si}_2\text{Te}_6$ (black, behind the Cernox). The thin gold wires are electrical leads electrically attached to the sample and Cernox with an EPO-TEK H20E epoxy (silver). The light brown background is GE varnish used to thermally anchor the Cernox and the sample. Note that $H \parallel c$ -axis and $I \parallel a$ -axis.

Fig. 12. Temperature change ΔT (K) as functions of applied current I (mA) and current density J (A/cm^2) at $T = 30$ K for representative magnetic fields $\mu_0 H \parallel c$ axis.

Fig. 13. Temperature change ΔT (K) as functions of applied current I (mA) and current density J (A/cm^2) at $\mu_0 H \parallel c$ axis = 3 T for representative temperatures T .

- c. *In short, Joule heating does exist but causes no more than 6K increase in sample temperature; its impact is therefore inconsequential for the phenomena reported in this manuscript (and Ref. 1).*
- d. *This discussion on Joule heating is particularly valuable. We have added it along with Figs. 11-13 to the main text (pages 4-5), Methods (pages 13-14) and Supplementary Figs. 4-5.*

In terms of the roles played by electric current and electric field (a good point made by the Reviewer), we can infer from the following data (Figs. 14-15) that the observed current-sensitive phenomena are primarily driven by electric current rather than electric field:

- a. *Our dielectric measurements using a Quan-Tech LCR meter reveal a large dissipation factor, DF, even at low temperatures, as shown in Fig.14. The large DF prevents a robust electric field from being established in this material, particularly at higher temperatures, say, 30 K at which most of our data is collected.*

Fig. 14. Dissipation factor, DF, as a function of frequency f up to 2 MHz at $T = 10$ K and $V = 1$ V. The separation between the two electrodes is approximately 0.3 mm.

- b. *We have measured a dozen samples with varying sample size for this study. Compiling these data, we have found a converging behavior in these samples: The critical magnetic field H_c (defined by the peak that occurs in the Hall resistivity ρ_{xy} (H) (see Fig.2 of the manuscript) essentially follows the same current-density J dependence shown in Fig. 15, which contains data obtained from four different samples. This indicates that the Hall response depends on applied currents.*

We have also added this discussion and Figs.14-15 to Methods and Supplementary Figs. 6-7.

Fig. 15. Critical Field $\mu_0 H_c$, as a function of current density J at $T = 30$ K for four different samples whose cross-sectional areas are displayed in the inset.

(2) The authors highlight that the Hall angle reaches a noteworthy value of 0.15. However, without a clear frame of reference, the significance of this value remains ambiguous. They compute the carrier density from the Hall resistivity and seemingly treat this resistance as ordinary Hall. It is worth noting, however, that an enlarged Hall angle for ordinary Hall resistance is not uncommon, in particular in materials with high carrier mobility.

Moreover, while the authors draw comparisons between the Hall angle in their study and those mentioned in Refs. 21-25, specifically papers 21-23 and 25 deal with the anomalous Hall effect. Such a comparison between the anomalous Hall effect and ordinary Hall resistance, seems incongruous. Although a Hall angle of 0.15 might be considered substantial for an anomalous Hall effect, its magnitude does not stand out as particularly exceptional.

To further elucidate their findings, the authors should provide additional clarity on the significance of the reported Hall angle, especially in relation to its broader implications and context within the literature.

This is a very good point. We have revised the manuscript accordingly (page 6 and Methods). In essence, σ_{xy}/σ_{xx} rapidly rises, reaching up to 0.15 and the estimated mobility of charge carriers μ is on the order of $100 \text{ cm}^2/\text{V}\cdot\text{s}$, one order of magnitude smaller than those reported [e.g., Refs. 19, 26]. In the revised manuscript we have modified the statement on this behavior.

Minor Comments:

(1) The current symbol "I" should be rendered either in italics or bold to prevent misunderstanding.

We have made these changes.

(2) As highlighted earlier, it is pivotal for the authors to share the sample dimensions. This would enable readers to more effectively gauge potential heating issues.

We have made these changes. The average sample size is stated on page 4.

Once again, we appreciate very much the Reviewers' thoughtful comments/suggestions, which have helped improve this manuscript significantly.

Reviewers' Comments:

Reviewer #1:

Remarks to the Author:

The authors have put considerable efforts into addressing the issues raised by the referees, and although I am not fully convinced by the CoC scenario, I think it is a promising explanation that deserves attention of the research community. Also given that the experimental observations by themselves are highly interesting, this work should be suitable for publication, after the authors address the following issues:

- (1) The field- and current-switching of the CoC should lead to clear signatures in neutron scattering, given that MCoC is estimated $\sim 0.1\mu\text{B}$. Can the authors describe expectations in neutron diffraction for the current-field phase diagram at selected temperatures, such as 30 K, 70 K, and 100 K? In particular, what are expected for the measured intensities at a ferromagnetic peak for different regimes in Fig. 2(a)-(e)? Can the authors make predictions that can be used to test the CoC scenario? These can then be tested by the authors or other workers interested in this material.
- (2) Please add more details in figure captions, for example in Figs. 1 a&b, explain what the triangle symbols are, and what the different colors correspond to.
- (3) Is the red line in Fig. 2f expected to cross zero at $H=0$ or $H>0$?
- (4) Figs. 3a&b, why is the crossover field independent of temperature?
- (5) Line 120, what is special about 3mA?
- (6) Fig. 2l, what are α_C and α_T ?

Reviewer #2:

Remarks to the Author:

I acknowledge and appreciate the authors' diligent efforts in addressing the concerns I previously raised. Their implementation of direct sample temperature measurements using mounted thermometers and their examination of current-induced phase transitions across various sample and current densities are particularly commendable. While their approach is noteworthy, some unresolved issues persist, such as the limited range of current density-dependence and the potential discrepancy between the thermometer and actual sample temperatures. Nonetheless, the observed current-induced phase transitions, even with minimal current changes, offer valuable insights both fundamentally and from an application point of view. Therefore, I believe that this manuscript is appropriate for publication in Nature Communications.

Regarding the heating aspects, I extend my gratitude to the authors for correcting my previous miscalculations by an order of magnitude. Nevertheless, there are certain assumptions in their methodology that I believe require reevaluation. In particular, the authors' choice of a 0.1 mm sample length, based on voltage lead distances, seems underestimated. A more accurate length would likely be around 1 mm, corresponding to current lead distances, which would better represent the actual zone of heating.

My reservations extend to specific points made in the authors' response, especially on pages 18-19:

- (a) The authors' interpretation of the anomalous increase in resistance with minimal current in the insulating state warrants further scrutiny. According to Reference 1, Figure 1a, there is a notable decrease in resistance below T_c for this material. As such, a rise in resistance could occur as heating brings the sample closer to T_c , which casts doubt on the authors' original argument.
- (b) The paper discusses a significant 2000% increase in resistance under a 7 T magnetic field with increased current. However, Reference 1, Figure 1a shows a marked change in resistance (by more than 100-fold) between 20 K and 100 K under the same magnetic field, especially near T_c . This implies that a rapid increase in resistance due to heating is not an unexpected phenomenon, thus weakening the authors' initial claim.

(c) The assertion that the current-induced phase transition is less significant in the H//a orientation is not entirely convincing. As shown in Figure 1b in Ref. 1, magnetoresistance is considerably lower in H//a compared to H//c. While the manuscript does not elaborate on the temperature dependence of resistance under magnetic fields for H//a, it is reasonable to infer that the variations would be less pronounced than in H//c. This observation aligns with the lower heating effect in H//a, challenging the authors' standpoint.

Despite these concerns, the direct measurement of the sample temperature strongly indicates minimal heating, which is a significant finding. Based on these observations and analyses, I recommend the publication of this manuscript in its current form.

**Authors' point-by-point response (bold) to comments/questions by the two reviewers
(References cited below follow the same numbering as in the revised manuscript)**

Response to Reviewer One

First of all, we are very thankful to the Reviewer for his recognition of the importance of this work and that this manuscript be approved after the questions below are addressed.

Reviewer #1 (Remarks to the Authors)

- (2) The field- and current-switching of the CoC should lead to clear signatures in neutron scattering, given that M_{CoC} is estimated $\sim 0.1\mu_B$. Can the authors describe expectations in neutron diffraction for the current-field phase diagram at selected temperatures, such as 30 K, 70 K, and 100 K? In particular, what are expected for the measured intensities at a ferromagnetic peak for different regimes in Fig. 2(a)-€? Can the authors make predictions that can be used to test the CoC scenario? These can then be tested by the authors or other workers interested in this material.

This is a good question. We have attempted to directly measure the COC-induced orbital moments M_{CoC} using neutron scattering at Oak Ridge National Lab. However, direct measurements of M_{CoC} are a daunting challenge for this particular material for at least two reasons:

- a. **$Mn_3Si_2Te_6$ is a ferrimagnet, thus the ferrimagnetic and nuclear peaks adversely overlap. In addition, M_{CoC} is on the order of $0.1 \mu_B$, one order of magnitude smaller than the Mn moments (Ref. 8). It is therefore particularly difficult to reliably measure M_{CoC} in a direct manner.**
- b. **A polarized neutron source would be helpful if the magnetic state is antiferromagnetic. However, in this case, the ferrimagnetic state inevitably depolarizes polarized neutrons.**

That said, we are exploring new approaches/techniques to delve deeper into the COC state.

- (2) Please add more details in figure captions, for example in Figs. 1 a&b, explain what the triangle symbols are, and what the different colors correspond to.

We appreciate the referee's careful reading. We have added the following to the Fig. 1 caption for clarification: *"the different colors indicate different magnitudes of ab-plane COC and M_{coc} ; the green triangles denote off-ab-plane COC that are insignificant [1]."* See page 10 of the manuscript.

(3) Is the red line in Fig. 2f expected to cross zero at $H=0$ or $H>0$?

The red line in Fig.2f represents the peak or critical field H_c of the Hall resistivity $\rho_{xy}(H)$ (Figs. 2a-2e) as a function of current I . Since a finite current must be applied for $\rho_{xy}(H)$, H must be finite, i.e., $H > 0$ in Fig. 2f.

(4) Figs. 3a&b, why is the crossover field independent of temperature?

The crossover is a broad region related to the existence of the COC. Since the COC is robust below T_c ($= 78$ K), no distinct temperature dependence of the crossover is discernible below T_c . The gray area is approximately defined, and mainly serves as a guide to the eye. We have added a note to the figure caption for clarity. See pages 7 and 11 of the manuscript.

(5) Line 120, what is special about 3mA?

As we state in the text pages 5-6 and in Fig. 2, $I = 3$ mA separates the COC state and trivial state: i.e., when $I > 3$ mA, the COC state is destroyed and the trivial state emerges, as shown in Fig. 2.

(6) Fig. 2l, what are α_C and α_T ?

We assume that the referee is referring to Fig. 3l. The parameter α denotes the exponent of the scaling relation, i.e., $\sigma_{xy} \propto \sigma_{xx}^\alpha$. α_c corresponds to the COC state, and α_T to the trivial state. We have added a note to the figure caption to clarify it. See pages 8 and 11 of the manuscript.

Response to Reviewer Two

We are grateful to the Reviewer for his/her approval of the manuscript by concluding, “*I recommend the publication of this manuscript in its current form.*”

We are happy to address the Reviewer’s questions concerning our previous response (page 18-19) and previous work reported in Ref. 1.

My reservations extend to specific points made in the authors' response, especially on pages 18-19:

(a) The authors' interpretation of the anomalous increase in resistance with minimal current in the insulating state warrants further scrutiny. According to Reference 1, Figure 1a, there is a notable decrease in resistance below T_c for this material. As such, a rise in resistance could occur as heating brings the sample closer to T_c , which casts doubt on the authors' original argument.

We are indeed mindful that Joule heating could cause spurious behavior, therefore we particularly appreciate the Reviewer’s questions related to this matter and our previous response regarding the bistable, first-order switching at 10 K (Fig. 10 in the previous response, or Fig. 4 in Ref.1).

The Reviewer correctly observes that there is a brief drop in the resistivity from $\sim 90 \Omega \text{ cm}$ near $T_c = 78 \text{ K}$ to $50 \Omega \text{ cm}$ near 70 K at $H = 0$, as shown in Fig. 1a of Ref 1 (we present it below as Fig .1 for convenience in this discussion). However, this drop does not have any worrisome implications for the bistable, first-order switching at 10 K presented in Fig. 10 in the previous response (Fig. 4 in Ref. 1), for the following four reasons:

1. As already demonstrated in our previous response (Fig. 13), Joule heating due to current densities $\leq 1.4 \text{ A/cm}^2$ ($I = 5 \text{ mA}$) could at most increase the sample temperature by a few degrees K. For the sake of discussion here we re-present the data as Fig. 2, below. Note that these data were collected following the Reviewer’s suggestion that a Cernox

thermometer (gold, front) be thermally contacted with a single-crystal sample of $\text{Mn}_3\text{Si}_2\text{Te}_6$ (black, behind the Cernox); here we re-introduce the image as Fig.3.

As seen, the Joule heating effect significantly weakens with increasing temperature; at $T \geq 50$ K (purple dots and line), the temperature increase is no more than 2 K for currents as high as 1.4 A/cm^2 or 5 mA, the highest current density or current applied in our study. The role of Joule heating in this T-regime is therefore minimal.

2. Also as discussed in our previous response, the bistable switching at 10 K (Fig. 4 of Ref. 1) leads to a first-order jump to a much higher value of voltage V or resistance R with increasing I . (Here, we re-present Fig. 4 of Ref. 1 as Fig. 4 in this response.) If this switching were due to Joule heating, V or R would decrease rather than increase because R at $H = 0$ follows an insulating behavior (i.e., the higher temperature corresponds to lower resistance) despite the brief drop in the resistivity near T_c (Fig.1).
3. Moreover, the switching is strictly between two values of V , thus the signature of the bistable switching is basically independent of applied currents (this is particularly true in the data in Fig. 4b below). Please also note that the switching or the jump in V , $\Delta V = 0.52 \text{ V}$, is triggered by a *tiny increase* of applied I , $\Delta I = 0.005 \text{ mA}$ ($= 2.035 - 2.030 \text{ mA}$).
4. In contrast, the resistivity decreases with increasing temperature is a gradual, continuous change (see Fig. 1 below); Joule heating cannot generate switching via a first-order transition between two reproducible values of V or resistivity, independent of applied currents (see Fig. 4 below).

In addition, we would like to mention that, via private communications, the results reported in Ref. 1 have been reproduced by another group, whose work is expected to be published soon.

Fig.1. The temperature dependence of the a -axis resistivity ρ_a at various magnetic fields [Ref.1].

Fig. 2. Temperature change ΔT (K) as functions of applied current I (mA) and current density J (A/cm^2) at $\mu_0 H_{\parallel c} = 3 \text{ T}$ for representative temperatures T .

Fig. 3. Joule heating measurements: A Cernox thermometer (gold, front) is thermally contacted with a single-crystal sample of $\text{Mn}_3\text{Si}_2\text{Te}_6$ (black, behind the Cernox). The thin gold wires are attached to the sample and Cernox with an EPO-TEK H20E silver epoxy. The light brown background is GE varnish used to thermally anchor the Cernox to the sample. Note that $H \parallel c$ -axis and $I \parallel a$ -axis.

Article

Fig. 4. Time-dependent bistable switching: The a -axis voltage V_a as a function of time t at $T = 10$ K for **(a)** $H = 0$, **(b)** $\mu_0 H_{||c} = 7$ T and **(c)** $\mu_0 H_{||a} = 7$ T [Ref.1].

(b) The paper discusses a significant 2000% increase in resistance under a 7 T magnetic field with increased current. However, Reference 1, Figure 1a shows a marked change in resistance (by more than 100-fold) between 20 K and 100 K under the same magnetic field, especially near T_c . This implies that a rapid increase in resistance due to heating is not an unexpected phenomenon, thus weakening the authors' initial claim.

We respectfully disagree with this argument because of the following reasons similar to those presented above:

1. As shown in Fig. 2, Joule heating cannot cause a temperature increase of more than a few K. At 7 T this effect is even weaker, leading to a 2 K-increase at the highest current applied in this study. Here we recall another set of data in our previous response (as Fig. 5): At 7 T and 30K, the highest current applied, 5 mA, cannot cause more than an increase of 2 K, and the currents used for the data in Fig. 4b in question are less than 5 mA. More importantly, the 2000% jump is caused by a tiny increase in current by 0.01 mA (= 3.98-3.97 mA); see the purple and blue lines in Fig. 4b. Below 3.97 mA, the

voltage V remains unchanged, but a tiny increase by 0.01 mA triggers an abrupt jump in V . The values of V of the jump are independent of current, i.e., there are only two values of V for all currents applied in Fig. 4b.

2. In contrast, when H is applied along the a axis, the first-order bistable switching disappears. The value of the voltage V now sensitively depends on the value of applied current I , consistent with the behavior anticipated for a normal state, and Ohmic behavior is nearly recovered, as shown in Fig. 4c. Such an anisotropic behavior cannot be due to Joule heating, which is expected to be gradual, isotropic.
3. Again, the resistivity gradually, continuously changes with increasing T , as shown in Fig.1, which cannot reproduce the switching that accompanies a first-order transition between only two values of V or resistivity, independent of applied currents (see Fig. 4b).

Fig. 5. Temperature change ΔT (K) as functions of applied current I (mA) and current density J (A/cm^2) at $T = 30$ K for representative magnetic fields $\mu_0 H \parallel c$ axis.

(c) The assertion that the current-induced phase transition is less significant in the $H//a$ orientation is not entirely convincing. As shown in Figure 1b in Ref. 1, magnetoresistance is considerably lower in $H//a$ compared to $H//c$. While the manuscript does not elaborate on the temperature dependence of resistance under magnetic fields for $H//a$, it is reasonable to infer that the variations would be less pronounced than in $H//c$. This observation aligns with the lower heating effect in $H//a$, challenging the authors' standpoint.

The Reviewer correctly observes that the magnetoresistance for $H \parallel a$ is much smaller, only 20% at 14 T, as a result, the resistivity for $H \parallel a$ is 7-orders of magnitude greater than that for $H \parallel c$ when $H > 3$ T, as shown in Fig. 6 below (the mentioned Fig. 1b in Ref. 1). If Joule heating plays a significant role, then the Hall effect and other current-sensitive properties should accordingly show a much stronger current dependence, because power due to Joule heating $P = I^2 \cdot R$, where I is held constant, and R is 7-orders of magnitude greater than that for $H \parallel c$, would be 7-orders of magnitude stronger.

However, the experimental observations indicate otherwise: The change in V is much smaller, and behaves nearly normally (Fig. 4c), the Hall effect is much weaker and almost featureless, as shown in Fig. 1g of the manuscript (Note that a axis = b axis), etc. – There is no clear current sensitivity in the physical properties for $H \parallel a$, where Joule heating, if exists, should be significant and strong.

These observations effectively further rule out any significant role of Joule heating in our work.

Fig. 6. The magnetic field dependence of the a -axis magnetoresistance ratio $[\rho_a(H) - \rho_a(0)]/\rho_a(0)$ and the magnetization M (dashed lines, right scale) for $H \parallel c$ -axis (blue) and $H \parallel a$ -axis (red) [1].

Reviewers' Comments:

Reviewer #1:

Remarks to the Author:

The authors have satisfactorily addressed my concerns, and I recommend the revised manuscript for publication.

Reviewer #2:

Remarks to the Author:

All of my concerns have been properly addressed, and therefore I recommend the manuscript for publication in Nature Communications.